# A SET domain-containing protein and HCF-1 maintain transgenerational epigenetic memory

Chenming Zeng[1,2,8], Giulia Furlan[2,8], Miguel Vasconcelos Almeida [1,2,8], Juan C. Rueda-Silva [1,2,3], Jonathan Price[1,2], Helena Santos-Rosa[1,2], Jingxiu Xu[1], Yan Kuang[4], Enric Cata Socias[1], Jonas Mars [2], Pedro Rebelo-Guiomar [1], Meng Huang[4], Shouhong Guang [4], Falk Butter [5,6] & Eric A. Miska [1,2,3,7] ✉

Transgenerational epigenetic inheritance (TEI) allows epigenetic information to pass across generations through mechanisms such as small RNAs and histone modifications. Histone methylation is often deposited by SET domain-containing methyltransferases. Some SET proteins lack catalytic activity but still regulate chromatin and gene expression. Here, we characterize SET-24, a catalytically inactive SET domain protein that localizes to germline nuclei and is essential for germline immortality in *Caenorhabditis elegans*. In *set-24* mutants, small RNA-mediated epigenetic silencing is impaired. Proteomic, yeast two-hybrid, and pull-down assays show that SET-24 interacts with HCF-1, a chromatin factor linked to complexes like COMPASS, which deposits H3K4me3. Loss of SET-24 leads to increased H3K4me3 at transcription start sites of hundreds of genes. Although transcription remains largely unchanged, small RNA production is disrupted for about 30% of these genes. We propose that SET-24 preserves germline epigenetic memory by sustaining a chromatin environment that supports small RNA biogenesis across generations.

RNA interference (RNAi) is a highly conserved gene silencing mechanism present in a wide array of organisms, safeguarding genome integrity and modulating gene expression[1,2]. This process can be triggered by both exogenous small interfering RNAs (exo-siRNAs) and endogenous small RNAs, including endo-siRNAs and Piwi-interacting RNAs (piRNAs). In the nematode *Caenorhabditis elegans*, RNAi involves the synthesis of primary siRNAs, which cleave their target mRNAs within the cytoplasm. This initiates the amplification of secondary siRNAs, also known as 22G-RNAs, bound to Argonaute proteins, thereby amplifying the signal and leading to post-transcriptional gene silencing[3]. Additionally, within the nucleus, small RNAs and Argonaute proteins can promote the deposition of histone modifications such as

H3K9me3 and H3K27me3, resulting in transcriptional gene silencing of target genes[4–8].

Both exogenous and endogenous RNAi cause heritable gene silencing. For example, the silencing effects initiated by GFP-targeting double-stranded RNAs (dsRNAs) can last for multiple generations[9,10]. Argonaute-bound siRNAs and histone modifications such as H3K9me3, H3K23me3, and H3K27me3 have been implicated in the transmission of silencing across generations, a phenomenon also known as transgenerational epigenetic inheritance (TEI)[4,6,7,10–16]. Numerous factors related to histone modifications, such as H3K27me3 demethylation factors, the PRC2 complex, the putative H3K9 methyltransferases MET-2, SET-25, and SET-32, as well as SET-

[1]Department of Biochemistry, University of Cambridge, Cambridge CB2, United Kingdom. [2]The Gurdon Institute, University of Cambridge, Cambridge CB2, United Kingdom. [3]Department of Genetics, University of Cambridge, Cambridge CB2, United Kingdom. [4]School of Life Science, University of Science and Technology of China, Hefei, Anhui, China. [5]Institute of Molecular Biology, Mainz, Germany. [6]Institute of Molecular Virology and Cell Biology, Friedrich-Loeffler-Institute, Greifswald, Germany. [7]Wellcome Sanger Institute, Wellcome Trust Genome Campus, Cambridge, United Kingdom. [8]These authors contributed equally: Chenming Zeng, Giulia Furlan, Miguel Vasconcelos Almeida. ✉e-mail: eam29@cam.ac.uk

21, which methylates H3K23 in conjunction with SET-32, are involved in TEI[12,17–24].

In transgenic *C. elegans* with a *gfp::h2b* sequence controlled by a germline-specific promoter, exposure to *gfp* dsRNA results in the silencing of GFP, which is typically inherited for multiple generations[9]. Previous research has proposed three distinct steps to small RNA-driven TEI: initiation, establishment, and maintenance[18]. The establishment of TEI relies on essential chromatin modifiers, specifically SET-25 and SET-32[18]. Other factors, such as HRDE-1, are involved in both the establishment and maintenance of TEI[25–27].

The mortal germline (Mrt) phenotype in *C. elegans* is a heritable trait characterized by progressive sterility across successive generations[28,29]. Over the past decades, numerous Mrt mutants have been identified, many of which are associated with RNAi and TEI[27]. For instance, mutations in the WAGO-4/ZNFX-1 complex, SET-25, and SET-32, and some nuclear RNAi pathway factors such as HRDE-1, lead to Mrt phenotype in *C. elegans* grown at 25 °C[17–19,25,30–32]. It has been hypothesized that RNAi machinery might promote germline immortality by establishing an epigenome conducive to germ cell quiescence[30]. However, the underlying mechanisms for defects in nuclear RNAi or TEI, simultaneous to temperature-sensitive Mrt phenotype, remain unclear. For piRNA-driven silencing, aberrant silencing of histone genes in piRNA mutants was proposed to underlie their Mrt defect[33].

Wild isolates of *C. elegans* offer a rich source of natural genetic variation for investigating worm behaviours, phenotypes, and their underlying mechanisms. Some wild isolates exhibit a temperature-sensitive Mrt phenotype at 25 °C[34]. In a previous study, a deletion of the *set-24* gene was identified in the wild *C. elegans* isolate MY10, which displayed a Mrt phenotype[35]. SET-24 encodes a protein that contains a conserved SET domain, originally identified from Su(var)3-9, Enhancer of zeste, and Trithorax histone methyltransferases (HMTs)[36]. SET domain-containing proteins are a well-characterized group of histone-modifying factors that play a critical role in gene regulation processes. Dysfunction of these factors is frequently associated with diseases, including cancer, developmental disorders, and aging-related pathologies[37]. Unlike the well-established functions of *C. elegans* SET-25 and SET-32 in H3K9 methylation and TEI, the function and mechanism of SET-24 remain elusive.

In this study, we investigated the function of SET-24. The SET domain of SET-24 shares similarity with the catalytically inactive SET domain of MLL5 in *Homo sapiens* (HsMLL5), which belongs to the *Saccharomyces cerevisiae* (ScSET3) subfamily. Accordingly, in vitro methyltransferase assays indicated that SET-24 does not show apparent methyltransferase activity. Animals with deletions in the coding sequence of *set-24* exhibit a Mrt phenotype at 25 °C, indicating that SET-24 is required to maintain germline immortality across generations. GFP RNAi inheritance assay demonstrated that SET-24 plays a role in the maintenance of transgenerational gene silencing. We also found that SET-24 is a germline-specific factor. We identified HCF-1 as a direct interacting partner of SET-24, and the two proteins colocalize in the nucleus. Although HCF-1 also regulates TEI, its depletion produces enhanced RNAi inheritance, the opposite phenotype to SET-24 depletion. Through chromatin and small RNA profiling, we found that SET-24 is required to regulate H3K4me3 and 22G-RNA levels in a subset of genes targeted by small RNA pathways. In summary, our findings reveal that SET-24, a factor with a catalytically inactive SET domain interacts with HCF-1 and is required at the interface between chromatin modifications and small RNA production, maintaining TEI.

## Results

### SET-24 is a member of the ScSet3 SET subfamily and lacks histone methyltransferase activity

The *C. elegans* genome encodes dozens of SET domain-containing proteins[20,38,39]. By aligning the SET domain sequences of these proteins and constructing a phylogenetic tree, we found that the SET domain of SET-24 is most similar to those of SET-9 and SET-26, paralogs with over 90% sequence similarity[40] (Fig. 1a). The SET domains of SET-9 and SET-26 are likely homologous to the SET domains of SET3 in *Saccharomyces cerevisiae* (ScSET3), UpSET in *Drosophila melanogaster* (DmUpSET), and MLL5 in *Homo sapiens* (HsMLL5), all of which belong to the ScSet3 subfamily of SET domains[41–43]. We also aligned the SET domain sequence of SET-24 with those of human proteins containing SET domains. Among all these proteins, the SET domain of SET-24 shares the highest similarity with those of HsMLL5 and HsSETD5 (Fig. 1b).

Members of the ScSet3 subfamily are generally considered catalytically inactive in regard to methyltransferase activity[42]. To further investigate this, we aligned the sequence of SET-24's SET domain with archetypal active and inactive SET domain-containing proteins. Like the ScSet3 subfamily members, the SET domain of SET-24 lacks several key residues that are crucial for catalytic activity (reviewed in refs. 42,44,45). For instance, the asparagine-histidine (NH) motif, essential for hydrogen bonding of the methyl donor SAM and the tyrosine (Y) motif, which functions by positioning target lysine in the catalytic site, are absent in SET-24 (Fig. 1c). These similarities to the ScSet3 subfamily and the absence of conserved motifs indicate that SET-24 is a member of the ScSet3 subfamily. To further validate this, we tested the methyltransferase activity of SET-24 on histones in vitro using a non-radioactive methyltransferase assay (Fig. 1d). We purified MBP-tagged full-length *C. elegans* SET-24 (CeSET-24) and *S. cerevisiae* SET2 (ScSET2), an active H3K36me3 methyltransferase, as a positive control[46,47] (Supplementary Fig. 1a). While the assay confirmed the methyltransferase activity of ScSET2, SET-24 did not exhibit apparent activity on any of the four core histones, indicating that SET-24 lacks histone methyltransferase activity (Fig. 1e). Besides SET-9/24/26, three other SET domains, of SET-5/22/28, lack catalytic residues and are therefore also members of the ScSet3 subfamily (Supplementary Fig. 1b). This suggests that the SET domains in the branch highlighted in Fig. 1a are likely catalytically inactive.

Most members of the ScSet3 subfamily possess a conserved PHD domain and a SET domain within their sequences[48] (Supplementary Fig. 2a). While the SET-24 sequence lacks the PHD domain, it contains two SPK domains (associated with SET, PHD, and Protein Kinase) of unknown function[49] (Supplementary Fig. 2a). Proteins containing SPK domains are predominantly found in nematodes, with some present in *C. elegans* (Supplementary Fig. 2b). We used AlphaFold3 to predict the structures of the SPK domains (SPK1 and SPK2) in SET-24 and conducted a search for similar structures with Foldseek. The folds of the SET-24 SPK domains are very similar to the MYB-like domains of the telomere-binding proteins *C. elegans* TEBP-1 and TEBP-2, which can bind directly to double-stranded telomeric DNA sequences using their third MYB-like domain[50,51] (Supplementary Fig. 2c).

### *Set-24* mutants display germline abnormalities and occasional escape from sterility

To explore the role of SET-24 in *C. elegans*, we first characterized the phenotypic impact of *set-24* mutations. To address this, we created two alleles in a wild-type N2 background: *set-24(mj617)*, an allele with a deletion of the entire coding sequence of *set-24* and *set-24(syb7014)*, a recreation of the *set-24* allele *mf123* from a previously described wild isolate that encodes a truncated version of the protein due to an early STOP codon after the 188th amino acid[35] (Supplementary Fig. 3a). When grown at 25 °C, the reference N2 strain could survive and usually be maintained for an indefinite number of generations, while the nuclear RNAi-defective mutant *hrde-1* worms became sterile after 1 to 8 generations, showing a Mrt phenotype[25]. Similarly, when *set-24(syb7014)* and *set-24(mj617)* mutants were cultured at 25 °C, the worms exhibited the Mrt phenotype and became sterile after 1 to 8 generations (Fig. 2a, b). Remarkably, certain *set-24* lines escaped the Mrt phenotype and remained fertile at 25 °C for more than 20 generations. We referred to these lineages as "escapees" (Fig. 2a, b).

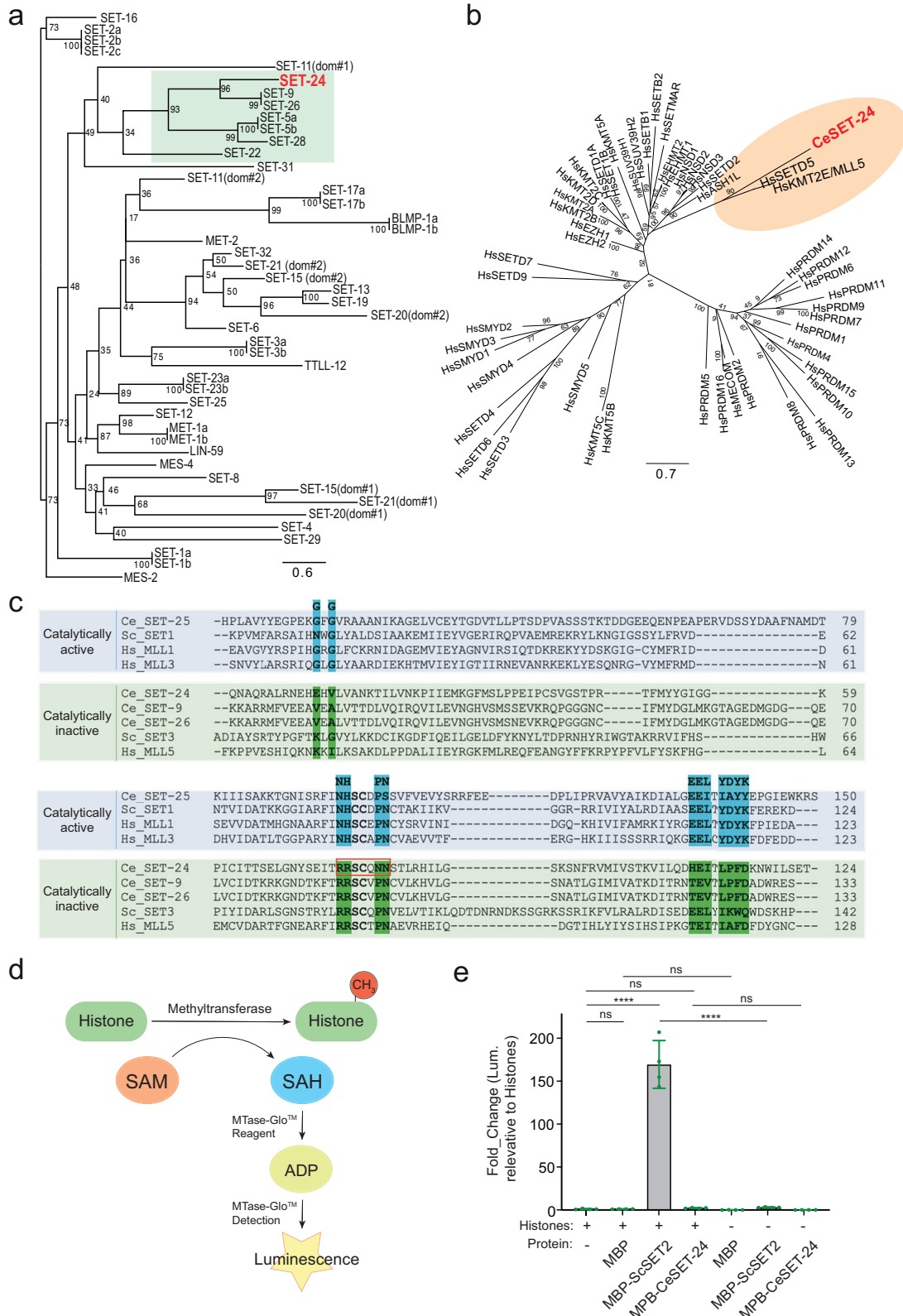

After growing the worms at 25 °C for several generations (G1-G3, from a parental P0), the germline of *set-24* mutant strains displayed abnormalities, such as a short or atrophied gonad, or even an empty germline, which were not observed in the germline of wild-type worms. The percentage of *set-24* mutant worms with abnormal germline increased from the parental P0 to G3. However, beyond G4, most *set-24* mutant worms lack any observable germline defects (Figs. 2c, and Supplementary 3b, c), likely because the "escapees" became the dominant population and remained fertile in later generations. In conclusion, *set-24* mutants show a Mrt phenotype that is associated with defects in germline development, but some *set-24* lines escape sterility.

**Fig. 1 | The SET domain of SET-24 is a member of the ScSet3 SET subfamily and is catalytically inactive. a** Maximum likelihood phylogenetic tree comparing the protein sequences of SET domains of selected SET domain-containing genes in *C. elegans*. Values next to the tree nodes are branch supports, calculated with 1000 ultrafast bootstrap replicates. **b** Phylogenetic tree of human SET domains and the SET domain of *C. elegans* SET-24. Values next to the tree nodes are branch supports, calculated with 1000 ultrafast bootstrap replicates. **c** Protein sequence alignment of the *C. elegans* SET-24 SET domain with SET domains from catalytically inactive Set3 SET subfamily and catalytically active CeSET-25, ScSET1, HsMLL1, and HsMLL3. The residues important for catalytic activity are highlighted. Ce, *Caenorhabditis*

*elegans*; Hs, *Homo sapiens*; Sc, *Saccharomyces cerevisiae*. **d** Schematic representation of the bioluminescent assay to measure methyltransferase activity. **e** Bar graph showing the in vitro methyltransferase activity of recombinant MBP-ScSET2 (SET2 from *Saccharomyces cerevisiae*) and MBP-CeSET-24 (SET-24 from *Caenorhabditis elegans*) full-length proteins. Reactions with 26 mM MBP, 20 mM MBP-ScSET2, 30 mM MBP-CeSET-24, and 1 µg histones. Error bars represent mean ± SD. *n* = 4 independent replicates. The corresponding data points are shown. Statistical significance was assessed with two-way ANOVA multiple comparisons. ns, not significant; ****$p$ value < 0.0001. MBP, Maltose-Binding protein. Source data and exact $p$ values are provided as a Source Data file.

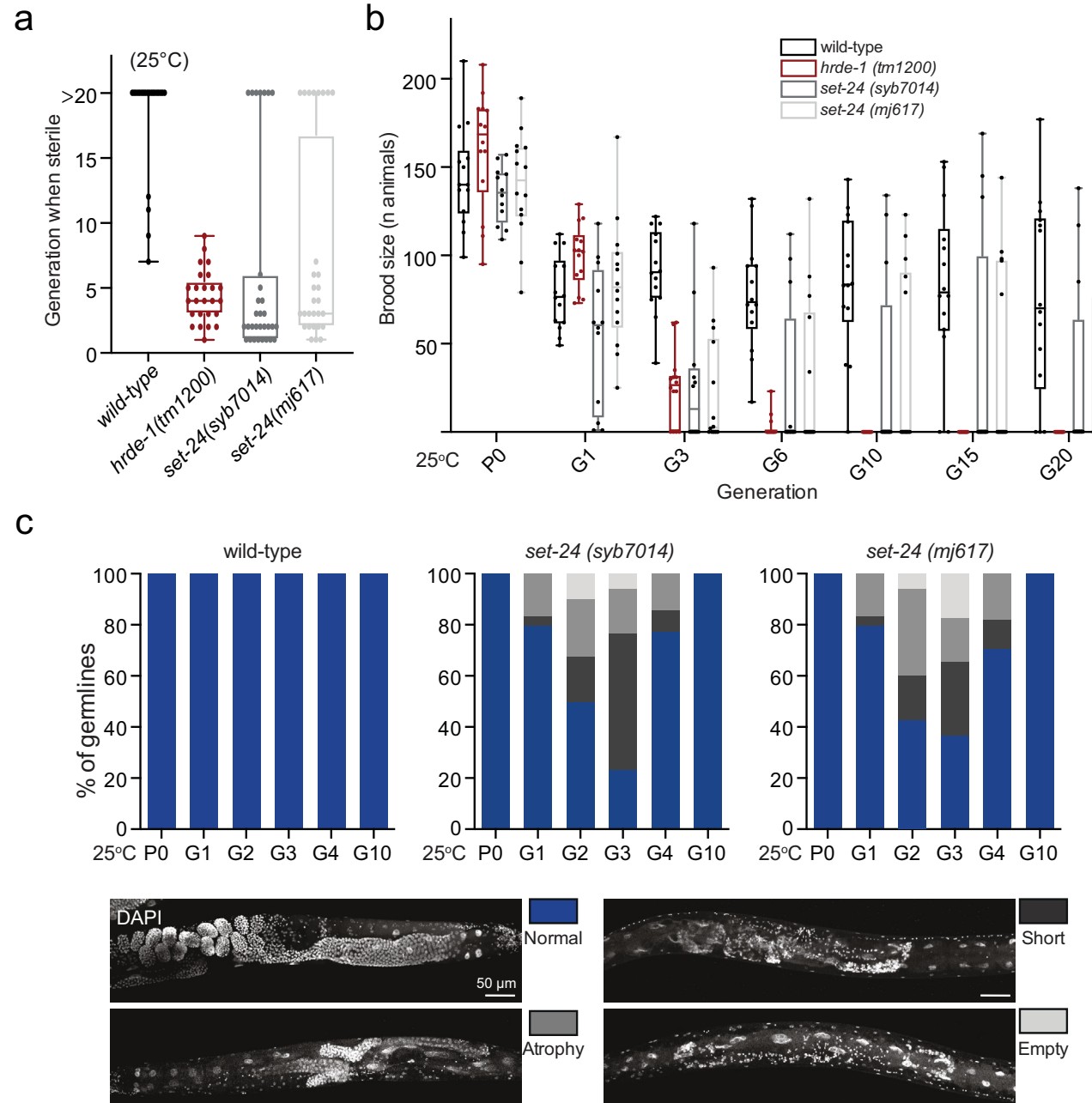

**Fig. 2 | SET-24 is required for germline integrity and fertility. a, b** *set-24* mutants are germline mortal (Mrt) at 25 °C. **a** Boxplots showing the number of generations elapsed until animals become sterile. *wild-type*, *n* = 21 worms; *hrde-1(tm1200)*, *n* = 24 worms; *set-24(syb7014)*, *n* = 28 worms; *set-24(mj617)*, *n* = 31 worms; **b** quantification of the number of progeny over generations, *n* = 15 per genotype. Box whiskers represent maxima and minima, center lines show median, and box

bounds show the quartiles. **c** *set-24* mutants display progressive germline degeneration under heat-stress. Quantification of the proportions of normal, short, atrophic, and empty germlines (*n* > 30 worms per generation). Results shown in (**a**–**c**) came from worms collected at different times. Source data are provided as a Source Data file.

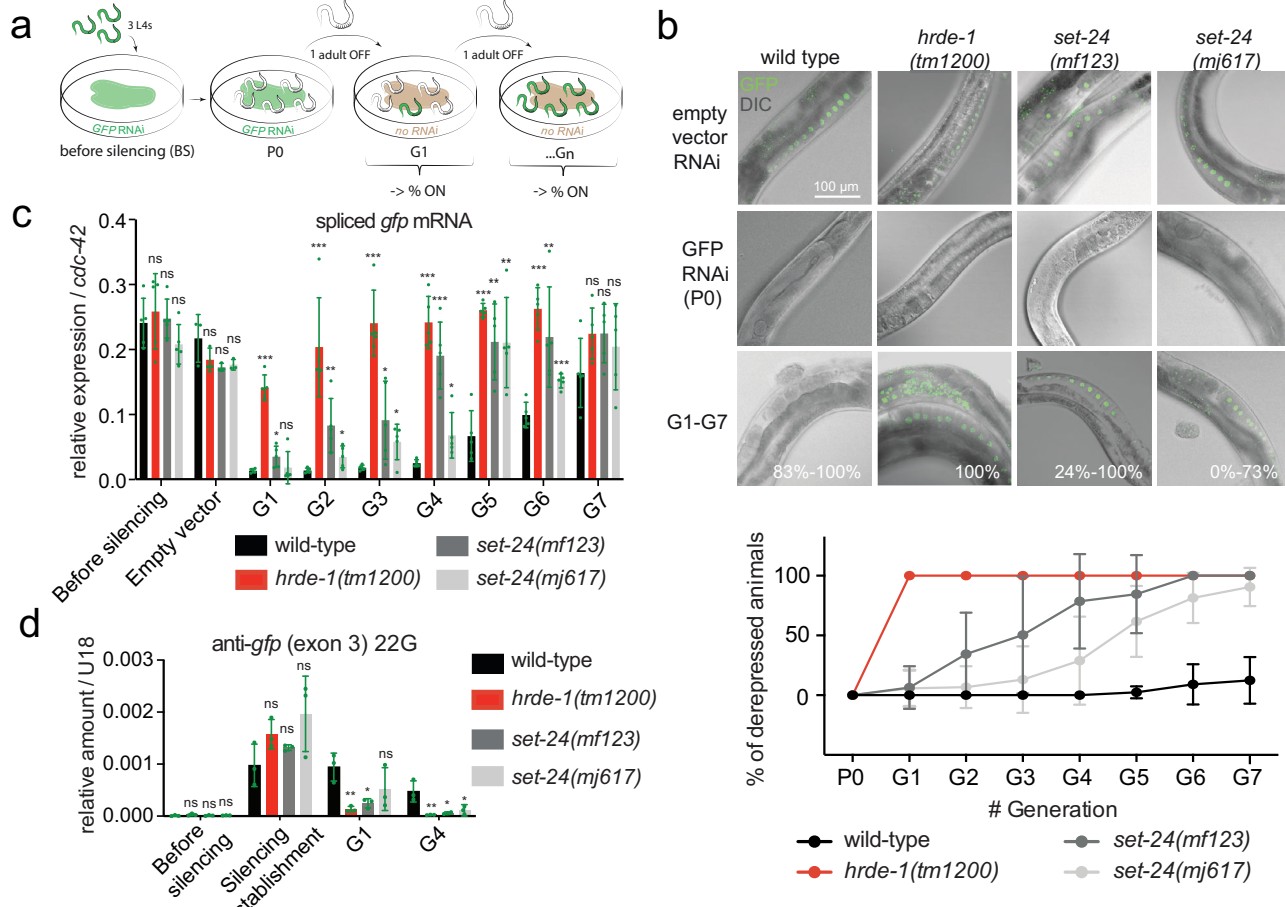

**Fig. 3 | SET-24 is required for RNAi inheritance. a** Schematic representation of the experimental procedure. **b** Top: Representative images of GFP transgene activity in *wild-type*, *hrde-1* and *set-24* mutant germlines. Bottom: Percentage of derepressed individuals in every generation. Means and standard deviations are shown. Countings on >20 animals, experiments on N = 5 animal lineages per genotype, *n* = 3 independent replicates. **c** RT-qPCR analysis of GFP transgene (spliced) mRNA expression at every generation. Means and standard deviations are shown, *n* = 5 independent replicates for the 'Before silencing' and 'G1–G7', and *n* = 3 independent replicates for the 'Empty vector'. **d** RT-qPCR measurement of the abundance of a representative anti-GFP 22G-RNA. Means and standard deviations are shown, *n* = 3 independent replicates. **c**, **d** Statistical significance was assessed with the two-sided unpaired *t*-test with Welch's correction. Stars indicate *p* value, ns, not significant; *0.01 ≤ *p* value < 0.05; **0.001 ≤ *p* value < 0.01; *** *p* value < 0.001. Comparisons were with wild-type strain. The corresponding data points are shown. Source data and exact *p* values are provided as a Source Data file.

## SET-24 is required to sustain heritable RNAi

Germline immortality is subject to genetic regulation. Previous studies have demonstrated that the absence of various factors associated with nuclear RNAi and TEI contributes to germline immortality[4,16,17,22,24–26,31,32,52–55]. Additionally, several SET domain-containing genes have been implicated in TEI[27]. Thus, we aimed to investigate if *set-24* mutants exhibit defects in RNAi inheritance. We explored SET-24's involvement in TEI using a previously described GFP::H2B transgenic strain[10] (Fig. 3a). In this strain, GFP is consistently expressed in germline and embryo nuclei. This transgene can be silenced by feeding worms with bacteria expressing dsRNAs targeting the *gfp* sequence. Even after removing the RNAi trigger, the silencing persists in subsequent generations[10] (Fig. 3b).

In worms with defective *hrde-1*, exposure to *gfp* dsRNA did not lead to heritable silencing of GFP::H2B, consistent with previous findings[25] (Fig. 3b). In *set-24* mutant strains, the silencing gradually decreased over a few generations after the removal of the RNAi bacteria (Fig. 3b). This decline was evaluated by determining the percentage of worms expressing GFP and measuring the relative expression level of spliced GFP mRNA using quantitative polymerase chain reaction (qPCR) (Fig. 3b, c). Given the roles of secondary siRNAs (22G-

RNAs) in heritable RNAi, we assessed the abundance of anti-GFP 22G-RNAs, which accumulate during the establishment of silencing. Anti-GFP siRNAs in *set-24* mutant worms did not persist at the same levels as in wild-type worms, one and four generations after the removal of the silencing trigger (Fig. 3d). These findings highlight the critical role of SET-24 in the maintenance of dsRNA-triggered TEI.

The piRNA pathway safeguards the germline against transposons, by recognizing their transcripts and eliciting 22G-RNA biogenesis, which drive target silencing[56–58]. The piRNA sensor strain expresses an *mcherry::his-58* fusion gene with a recognition site in the 3'UTR for an abundant piRNA[59]. In a wild-type background, the sensor is silenced, but depletion of factors involved in the piRNA pathway, such as *prg-1* and *hrde-1*, leads to its derepression. To investigate the potential role of *set-24* in the piRNA pathway, we crossed *set-24* mutants with the piRNA sensor strain. In contrast to the *hrde-1* mutant, the depletion of *set-24* did not derepress the piRNA sensor (Supplementary Figs. 4a, b). These findings suggest that *set-24* is dispensable for the initiation of piRNA-dependent silencing.

Overall, these results demonstrate that in the absence of SET-24, siRNA signals do not persist across generations and RNAi inheritance is impaired. Also, piRNA-dependent silencing is not affected by SET-24.

## SET-24 is a germline-specific factor and is localized to germline nuclei

Most TEI factors, such as the nuclear and perinuclear Argonautes, HRDE-1 and WAGO-4, respectively, are expressed in the germline[17,25,31,32]. This prompted the investigation of the expression pattern of SET-24. set-24 mRNAs have been previously detected in the C. elegans germline during the first larval stage[60]. In our RT-qPCR assay, set-24 mRNAs are predominantly detectable in embryo, L4, and adult stages, mirroring the mRNA expression pattern of the germline factor pie-1 (Supplementary Fig. 5a). set-24 mRNAs were not detected in glp-4(bn-2) worms grown at 25°C, which lack a germline at this restrictive temperature (Supplementary Fig. 5b). Moreover, set-24 mRNAs were still expressed in fem-1(hc17) and fog-2(q71) at the same temperature, despite the absence of sperm or oocytes respectively (Supplementary Fig. 5c). These mRNA expression patterns indicate that set-24 mRNA is confined to the C. elegans germline, both in spermatogenic and oogenic gonads.

Subsequently, we examined the localization of the SET-24 protein. We created an endogenously GFP-tagged SET-24 allele (Supplementary Fig. 3a). Similar to the wild-type strain, SET-24::GFP animals can survive and be maintained for numerous generations at 25 °C, suggesting that the GFP at the C-terminus of SET-24 does not alter its function (Supplementary Fig. 5d, e). SET-24::GFP expression is restricted to germline nuclei (Fig. 4a and Supplementary Fig. 5f). Notably, unlike the germline perinuclear factor PGL-3, expressed throughout the entire gonad from the mitotic region to the diakinesis region in oocytes (Supplementary Fig. 5g), SET-24::GFP was only expressed between the mitotic and pachytene regions, with a clear fade in signal at the boundary of the pachytene and diplotene regions (Fig. 4a). In embryos, SET-24::GFP was solely expressed in germline-lineage cells, localized mainly to the nucleus, and showed partial overlap with PGL-3 (Fig. 4a).

To further examine the expression of SET-24, we analysed its colocalization with the histone marker H2B::mCherry and the synaptonemal complex marker tagRFP::SYP-1 using confocal microscopy in whole worms (Fig. 4b, c). Interestingly, SET-24::GFP forms puncta in germline nuclei (Fig. 4b, c). While partial overlap between SET-24::GFP and H2B::mCherry was observed, likely on chromosomal regions, the large SET-24 puncta were distinct from chromosomes. (Fig. 4b, d). In summary, we conclude that SET-24 is a germline-specific nuclear factor, expressed up to the pachytene region that partially colocalizes with Histone H2B. We note that the pool of SET-24 protein localized to puncta does not seem to directly associate with chromosomes.

SET-24 contains a catalytically inactive SET domain (Fig. 1). To determine whether this domain influences the expression and localization of SET-24, we generated multiple mutations, including amino acid substitutions, in-frame and frame-shift deletions in the SET domain of set-24::gfp and set-24::3xflag strains (Fig. 4e and Supplementary Fig. 5h). Specifically, mj673 and mj677 introduce five amino acid substitutions in the region of the SET domain conserved throughout the ScSET3 subfamily (highlighted in the red frame in Fig. 1c; Fig. 4e and Supplementary Fig. 5h). The mj675 mutation deletes a sequence spanning both intronic and exonic regions, but remains in-frame, whereas mj674 and mj676 introduce frame-shift deletions in set-24::gfp and set-24::3xflag strains, respectively (Fig. 4e and Supplementary Fig. 5h).

Analysis of transcript levels of set-24 revealed that set-24(mj673)::gfp and set-24(mj677)::3xflag strains exhibit mRNA levels comparable to those of wild-type set-24::gfp and set-24::3xflag, while set-24(mj675)::3xflag displays reduced set-24 mRNA abundance, likely due to intron loss (Supplementary Fig. 5i). SET-24::GFP or SET-24::3xflag proteins were not detected in the in-frame deletion strain set-24(mj675)::3xflag or in the frame-shift mutants, set-24(mj674)::gfp and set-24(mj676)::3xfalg strains. Notably, although transcripts in set-24(mj673)::gfp and set-24(mj677)::3xflag are maintained at wild-type

levels, GFP and FLAG signals are undetectable (Fig. 4f, g). These results indicate that the SET domain, and in particular the conserved amino acids within the ScSET3 subfamily (highlight in the red frame in Fig. 1c), are critical for protein stability in vivo, even though the domain itself is catalytically inactive.

## SET-24 interacts with HCF-1 in vivo and in vitro

Given the nuclear localization of SET-24 and its partial colocalization with H2B (Fig. 4d), we inquired if SET-24 is able to bind to chromatin and interact with other chromatin factors in vivo. To investigate this, we performed a chromatin immunoprecipitation (ChIP) assay using both SET-24::3xFLAG and SET-24::GFP strains. These approaches yielded no detectable DNA enrichment (Supplementary Fig. 6a). This could be attributed to either lack of SET-24 binding to chromatin, or technical issues in our ChIP procedures.

Subsequently, we identified SET-24 binding partners through IP-MS (Immunoprecipitation and Mass Spectrometry) and Y2H (Yeast-two Hybrid) assays using young adult (YA) worms expressing SET-24::3xFLAG. Host Cell Factor 1 (HCF-1), the ortholog of human HCFC1 and HCFC2, was identified by the two assays (Fig. 5a, b, Supplementary Fig. 6b, Supplementary Data 1 and 2). Furthermore, a GST-pull down assay demonstrated that recombinant GST-tagged full-length SET-24 physically interacts with MBP-tagged full-length HCF-1 in vitro, confirming that the interaction between SET-24 and HCF-1 is direct (Fig. 5c and Supplementary Fig. 6c).

HCF-1 is a highly conserved chromatin adapter protein that recruits histone-modifying complexes to chromatin[61–63]. It has been reported that HCF-1 plays a role in longevity and stress resistance in C. elegans[63,64]. In both whole worms and embryos, HCF-1 is a nuclear factor expressed in the germline and soma, while SET-24 is restricted to the germline[64] (Fig. 4a, Supplementary Figs. 6d, e). In the gonad, HCF-1 does not show a decrease in signal at the boundary between the pachytene and diplotene regions, unlike SET-24::GFP (Supplementary Fig. 6f). In the germline nuclei, HCF-1::mKate2 also forms puncta and colocalizes with SET-24::GFP, with both factors showing partial overlap with BFP::H2B (Fig. 5d, e). Depletion of HCF-1 does not result in detectable alterations in the expression, localization, or puncta formation of SET-24. Reciprocally, depletion of SET-24 does not affect HCF-1 localization within the germline (Supplementary Figs. 6g, h). The data shows that SET-24 and HCF-1 directly interact and colocalize in germline nuclei, but do not affect each other's expression or localization.

In humans, HCF-1 interacts with MLL5 through HCF-1's Kelch domain and MLL5's HCF-1 binding motif (HBM)[65]. In C. elegans, the MLL5 homologs SET-9 and SET-26 contain the HBM, but the SET-24 protein does not encode a sequence identical to HBM[63] (Supplementary Fig. 2a). To identify the regions responsible for the HCF-1-SET-24 interaction, we purified MBP- and GST- tagged truncated forms of HCF-1 and SET-24, respectively (Fig. 5f, g). The H1 N-terminal fragment of HCF-1 (1-378aa), which primarily contains the Kelch domain, but not the C-terminal fragment (379–782aa), directly interacts with full-length SET-24 (Fig. 5h and Supplementary Fig. 7a). Conversely, all SET-24 constructs containing the N-terminal region (1–239aa), which includes the SET domain and its flanking sequences, interacted with the H1 fragment of HCF-1 (Fig. 5i, Supplementary Figs. 7b, c). To further map the HCF-1 binding site of SET-24, we focused on its 1-239aa N-terminal region containing the SET domain. We created one construct with the SET domain alone (fragment S7), and another version of SET-24 fragment S5 in which the SET domain was deleted in-frame (fragment S8; Fig. 5g). Interestingly, the HCF-1 fragment H1 interacted both with fragments S7 and S8 (Fig. 5j, k, Supplementary Figs. 7d–f). Together, these data suggest the HCF-1 Kelch domain directly interacts with the SET domain of SET-24, while the SET domain of SET-24 and its flanking sequences both contribute to the interaction with HCF-1.

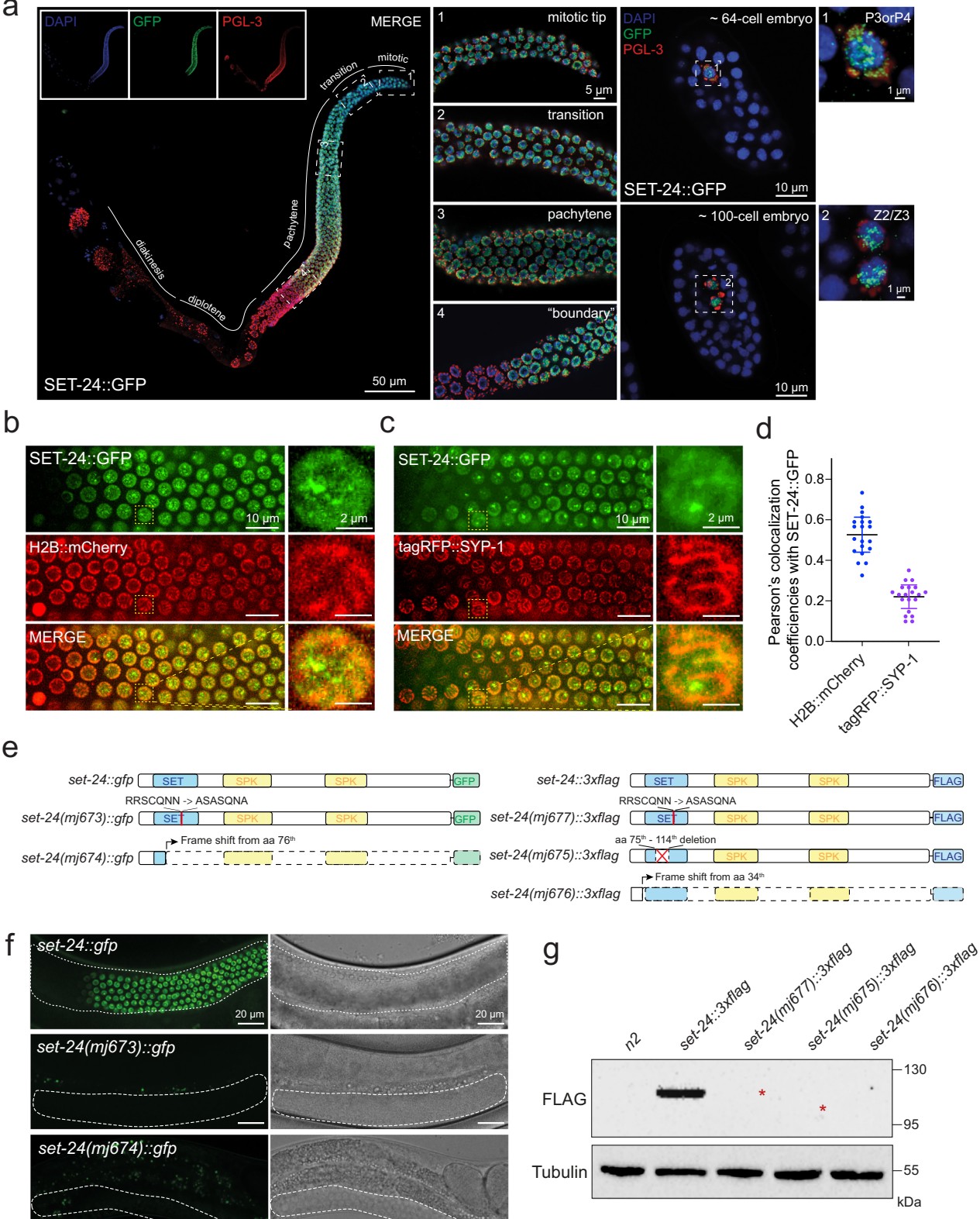

## HCF-1 inhibits the maintenance of heritable silencing

To investigate the roles of HCF-1, we obtained two alleles, *ok559* and *pk924*. The *ok559* allele carries an in-frame deletion in the Kelch domain, whereas *pk924* is a frameshift mutation[66] (Supplementary Fig. 6g). Both *hcf-1* mutants exhibited a reduced brood size at 20 °C and are nearly sterile at 25 °C within one or two generations, without producing "escapees", contrasting with the weaker phenotype of *set-24*

mutants (Fig. 6a). We next examined the brood size of *set-24; hcf-1* double mutants and found that their brood sizes were comparable to those of the *hcf-1* single mutants (Fig. 6a).

To assess if HCF-1 plays a role in TEI, we conducted a RNAi inheritance assay. Heterozygous *set-24* or *hcf-1* worms with a GFP::H2B transgene were fed *gfp* dsRNA, and the inheritance of silencing was monitored in subsequent generations (Fig. 6b). In this experimental

**Fig. 4 | SET-24 is a germline-specific factor and is localized to the nucleus. a** SET-24 localizes to germline nuclei. Representative immunofluorescence images of SET-24 localization (anti-GFP antibody, green) in dissected germlines of a SET-24::GFP strain. P-granules are stained with an anti-PGL-3 antibody, red, and DNA is stained with DAPI, blue. Numbered insets, zoom in on particular regions of the germline and on embryonic germ cells. *n* = 3 independently imaged worms with similar results. **b, c** Representative images of *C. elegans* expressing SET-24::GFP and H2B::mCherry (**b**) or tagRFP::SYP-1 (**c**) in whole worms. *n* = 10 independently imaged worms with similar results. **d** Pearson's colocalization coefficiencies between SET-24::GFP and H2B::mCherry or tagRFP::SYP-1 within a single nucleus, measured using ImageJ. Means and standard deviations are shown. *n* = 20 nuclei

from 10 different worms. Source data are provided as a Source Data file. **e** Schematic of deletions or mutations in *set-24::gfp* (left) and *set-24::3xflag* (right) strains. **f** Representative images of GFP expression and bright field in the indicated strains. The germline is outlined with white dashed lines. *n* = 10 independently imaged worms with similar results. **g** Western blot analysis of FLAG expression in the indicated strains using anti-FLAG antibody (F1804). Red stars indicate the expected sizes of *set-24(mj677)::3xflag* and *set-24(mj675)::3xflag*. The same membrane was re-probed with anti-tubulin antibody (ab6160). *n* = 3 independent experiments with similar results. The uncropped and unprocessed versions of blots are provided in Source Data.

setup, *set-24* mutants lost silencing faster than wild-type animals (Supplementary Figs. 8a, b), which is consistent with previous findings (Fig. 3b). In contrast, it takes more generations for *hcf-1* mutants to lose silencing compared to wild-type worms (Fig. 6c, d, Supplementary Fig. 8b). These results indicate that HCF-1 inhibits the maintenance of heritable silencing.

Next, heterozygous +/*set-24*; +/*hcf-1* worms carrying the GFP::H2B transgene were fed *gfp* dsRNA and the inheritance of GFP silencing was monitored in progeny with different genotypes (either wild-type, *set-24* or *hcf-1* single mutants, and *set-24; hcf-1* double mutants; Fig. 6e). Consistent with prior observations (Figs. 3b and 6b-d, Supplementary Figs. 8a, b), silencing was lost in fewer generations in *set-24* single mutants and was maintained for more generations in *hcf-1* single mutants (Fig. 6f, g). Importantly, the RNAi inheritance in *set-24; hcf-1* double mutants phenocopied *hcf-1* single mutants (Fig. 6f, g). To further validate these findings, heterozygous +/*set-24*; *hcf-1/hcf-1* or *set-24/set-24*; +/*hcf-1* worms carrying the GFP::H2B transgene were silenced, and the single mutants were compared with *set-24; hcf-1* double mutants (Supplementary Figs. 8c-f). Again, *set-24; hcf-1* double mutants maintain silencing similar to *hcf-1* single mutants, and for more generations than *set-24* single mutants (Supplementary Figs. 8c-f), corroborating previous results (Fig. 6f, g).

Together, these results indicate that HCF-1 and SET-24 both mediate TEI but display opposite inheritance phenotypes, with HCF-1 acting epistatically to SET-24 in this process.

## SET-24 is required to modulate H3K4me3 and small RNA levels
SET domain proteins are known to influence histone modifications[67]. Human HCF-1 interacts with mixed-lineage leukemia (MLL) and the H3K4 methyltransferase Set1[62,68,69]. In *C. elegans*, HCF-1 associates with the COMPASS complex, where the H3K4 methyltransferase SET-2 serves as an essential component[63,70,71]. To investigate if the depletion of *set-24* affects the global levels of histone modifications, we conducted a western blot assay using antibodies specific to various histone methylation and acetylation marks in YA worms. This assay was performed on worms either grown at 20 °C or for three generations at 25 °C. There were no observable changes in *set-24* mutants compared to wild-type worms (Supplementary Fig. 9a, b).

Since western blotting lacks the sensitivity to detect local changes in histone modification levels, we performed a ChIP-seq assay using an H3K4me3 antibody on both wild-type and *set-24* mutant YA worms grown at 20 °C. Overall, the H3K4me3 peak profiles were highly similar between strains (Supplementary Fig. 10a). However, ~10% of peaks showed increased H3K4me3 signal in *set-24(syb7014)* mutants, and most of these peaks overlapped with protein-coding genes (Supplementary Figs. 10b, c). In both strains, H3K4me3 was enriched at transcription start sites (TSS) of protein-coding genes (Fig. 7a), consistent with known patterns of this modification[72]. Importantly, the increase in H3K4me3 was not genome-wide but restricted to genes with TSSs already marked by H3K4me3 in the wild-types (Fig. 7b). We identified a subset of genes with significantly elevated H3K4me3 levels in *set-24(syb7014)* mutants, hereafter referred to as H3K4me3-enriched genes (Fig. 7a-c). Notably, the increase in H3K4me3 extended both

upstream and downstream of the TSS (Supplementary Fig. 10d). To assess if this was due to changes in nucleosome occupancy, we conducted histone H3 ChIP-seq and measured total H3 levels globally and at SET-24-regulated H3K4me3-enriched genes. H3 levels were unchanged between strains, indicating that the increase in H3K4me3 in *set-24(syb7014)* mutants reflects a true modification change rather than altered H3 abundance (Supplementary Figs. 10b, e). Cross-referencing with a published germline expression dataset revealed that ~90% of the 757 *set-24*-dependent H3K4me3-enriched genes are germline-expressed[73] (Supplementary Fig. 10f), further supporting a role for SET-24 in regulating H3K4me3 in the germline.

H3K4me3 is generally considered a histone mark associated with transcriptional activation[74]. To assess if the observed increase in H3K4me3 levels influences gene expression, we performed mRNA sequencing on wild-type and *set-24(syb7014)* mutant YA worms and analysed the transcriptional changes across all genes (Supplementary Data 3). We identified 12 genes with both H3K4me3 enrichment and mRNA upregulation in *set-24(syb7014)* mutants (Fig. 7d and Supplementary Data 4). These genes are also regulated by components of the COMPASS complex, SET-2 and WDR-5, as determined through a WormExp enrichment analysis[75] (Supplementary Data 5), further suggesting a role for SET-24 in H3K4me3 regulation through HCF-1 and its interaction with COMPASS. However, the majority of H3K4me3-enriched genes did not exhibit increased transcription in *set-24(syb7014)* mutants (Fig. 7d).

Given the role of SET-24 in the maintenance of heritable RNAi and inheritance of 22G-RNAs (Fig. 3b, d), we sought to investigate the potential link between SET-24 and 22G-RNA regulation. To do so, we sequenced small RNAs in wild-type and *set-24* mutants and analysed 22G-RNAs mapping to protein-coding genes (Supplementary Figs. 11a-c). We identified genes with significant changes in 22G-RNA expression in *set-24* mutants relative to wild-type (Supplementary Data 6) and classified them as *set-24*-regulated 22G-RNA targets. More such targets were identified in the *syb7014* allele than in the *mj617* allele, but the two datasets overlapped well (Supplementary Fig. 11c); therefore, subsequent analyses focused on *syb7014* for consistency with other sequencing datasets. 22G-RNAs can be sub-divided into distinct subpopulations, according to the Argonaute protein they associate with, the factors required for their biogenesis, and the genes targeted[76]. We found that the *set-24*-regulated 22G-RNA targets overlap with WAGO- and Mutator-regulated genes (Fig. 7e, Supplementary Fig. 11d, and Supplementary Data 7), suggesting SET-24 is required for silencing of a subset of endogenous WAGO and Mutator targets.

Because HCF-1 also regulates TEI, we analysed 22G-RNAs in *hcf-1(pk924)* mutants. 22G-RNA populations were altered *in hcf-1(pk924)* mutants (Supplementary Fig. 11b), and 55% of the identified *hcf-1*-regulated 22G-RNA targets showed germline expression, compared with 60% of *set-24*-regulated targets (Supplementary Fig. 11e). While some overlap was observed between the two datasets, most targets were distinct (Supplementary Fig. 11f). Although HCF-1 directly interacts with SET-24 and their depletion causes opposite phenotypes, the overlapping 22G-RNA targets did not consistently exhibit opposite

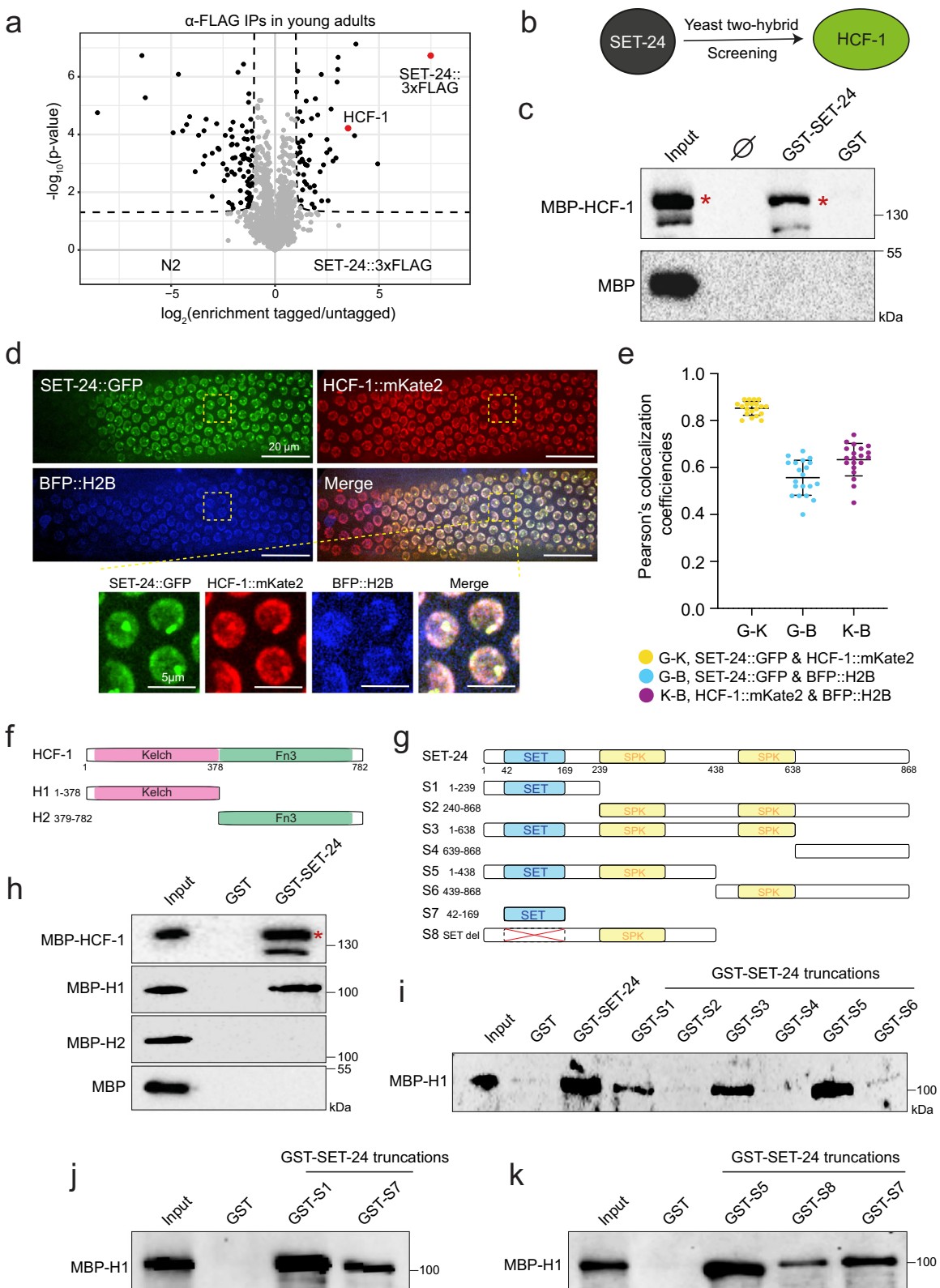

modes of regulation (Supplementary Fig. 11g). Thus, HCF-1 regulates 22G-RNA populations, including of a subset of targets shared with SET-24.

Next, we examined the correlation between H3K4me3 enrichment and *set-24*-regulated 22G-RNA targets (Fig. 7f). In total, 220 out of 757 H3K4me3-enriched genes in *set-24*(*syb7014*) mutants exhibited disrupted 22G-RNA levels (81 upregulated and 139 downregulated),

whereas only 13 out of 757 showed altered transcriptional levels (including 12 upregulated and 1 downregulated, Fig. 7d, f, Supplementary Data 4). Among the H3K4me3-enriched genes, few (e.-g.*Y39G8B.2* and *dyf-3*) exhibited both increased transcription and disrupted 22G-RNA expression (Supplementary Fig. 11h). However, a larger subset, including *C18B2.4* and *clec-24*, showed disrupted 22G-RNA levels without corresponding transcriptional changes (Fig. 7f, g).

**Fig. 5 | SET-24 directly interacts and colocalises with HCF-1. a, b** Identification of SET-24 interactors by immunoprecipitation followed by mass spectrometry (IP-MS, **a**) and Yeast-two Hybrid (Y2H, **b**). Statistical significance was assessed using a two-sided unpaired *t*-test. **c** In vitro GST-pulldown assays using GST-tagged full-length SET-24 or GST to bind MBP-tagged full-length HCF-1 or MBP. Bound proteins were analysed by Western blotting using anti-MBP antibody (ab49923). Red stars indicate the full-length MBP-HCF-1. GST, Glutathione S-transferase; MBP, Maltose-Binding Protein; ∅, empty lane. *n* = 3 independent experiments with similar results. **d** Representative images of *C. elegans* expressing SET-24::GFP, HCF-1::mKate2, and BFP::H2B in whole worms. *n* = 10 independently imaged worms with similar results. **e** Pearson's colocalization coefficiencies between SET-24::GFP, HCF-1::mKate2, and BFP::H2B within a single nucleus, measured using ImageJ. Means and standard deviations are shown. *n* = 20 nuclei from 10 different worms. Source data are provided as a Source Data file. **f, g** Schematic of MBP-tagged HCF-1 truncations (**f**) and GST-tagged SET-24 truncations (**g**). **h** In vitro GST-pulldown assays using GST-tagged full-length SET-24 or GST to bind MBP-tagged HCF-1 truncations or MBP. Bound proteins were analysed by Western blotting using anti-MBP antibody (ab49923). Red star indicates the full-length MBP-HCF-1. *n* = 3 independent experiments with similar results. **i, k** In vitro GST-pulldown assays using GST-tagged SET-24 truncations to bind the MBP-tagged H1 truncation of HCF-1. Bound proteins were analysed by Western blotting using anti-MBP antibody (ab49923). *n* = 3 independent experiments with similar results. The uncropped and unprocessed versions of blots for (**c** and **h**–**k**) are provided in Source Data.

These results suggest that SET-24 is dedicated to maintaining 22G-RNA levels.

Taken together, these results suggest that SET-24 modulates H3K4me3 and 22G-RNA levels in a subset of WAGO/mutator target genes.

## Discussion

In this work, we provide insights on the roles of SET-24, a catalytically inactive SET domain-containing factor expressed in germline nuclei. SET-24 is essential for maintaining germline integrity and heritable RNAi across generations. Intriguingly, SET-24 and its interactor HCF-1 are both required for the maintenance of heritable RNAi, yet their depletion produced opposing phenotypes. We will discuss below how the imbalance of H3K4me3 and 22G-RNA levels observed in *set-24* mutant animals may be relevant for the maintenance of epigenetic memory.

The SET domain superfamily consists of multiple subfamilies, one of which is the ScSET3 family. ScSET3 family members are characterized by similarities in domain structure, sequence, and biological function to the Set3 protein of *S. cerevisiae*[77]. Although these members are unlikely to possess catalytic methyltransferase activity, they play essential roles in various biological processes and are associated with epigenetic regulation[48,78]. However, mechanistic insight into the function of ScSET3 subfamily members is lacking. Notably, most ScSET3 family proteins, except for SETD5 in zebrafish, mouse, rat, and human, contain a PHD domain—a chromatin reader that recognizes methylated H3 lysine 4 (H3K4)[41,48,79,80]. Despite its role in the maintenance of H3K4me3 levels, SET-24 lacks a PHD domain and is therefore unlikely to recognize H3K4 directly. Some members of the ScSET3 subfamily regulate gene expression by recruiting histone deacetylases (HDACs) to specific genomic regions[48,63,77,81]. However, our IP-MS and yeast two-hybrid assays did not identify any HDACs as interacting partners of SET-24 (Supplementary Data 1 and 2), arguing against a direct interaction with HDAC complexes.

Unlike other ScSET3 subfamily members that possess PHD domains, SET-24 contains two SPK domains. The biological and biochemical functions of these SPK domains remain unclear. Interestingly, the SPK domains of SET-24 share structural similarity with MYB-like domains found in TEBP-1 and TEBP-2 (Supplementary Fig. 2c). One of three MYB-like domains in TEBP-1 and TEBP-2 directly binds to double-stranded telomeric DNA sequences[51]. Although we were unable to immunoprecipitate SET-24 together with chromatin, exploring the DNA-binding potential of the SET-24 SPK domains using alternative approaches could provide valuable insights. SPK domain-containing proteins are predominantly found in nematodes, with several identified in *C. elegans* (Supplementary Fig. 2b). We aligned SPK domains in *C. elegans* and found that the SPK2 domain in SET-24 is closely related to the SPK domain in OGR-2, which is known to influence progression of oogenesis in *C. elegans*[82] (Supplementary Fig. 2b). This similarity suggests that SET-24 might regulate germline development through its SPK domains. Interestingly, another SET domain-containing protein, SET-5, also contains an SPK domain, and its SET domain is very similar

to that of SET-24 and is likely catalytically inactive (Fig. 1a, Supplementary Figs. 1b, 2b). It will be important to determine if SET-24 and SET-5 have overlapping and/or redundant functions. The relatively small number of genes with associated H3K4me3 and 22G-RNA imbalance, may be explained by redundancy with other SET domain-containing proteins.

Additional SET domain-containing proteins highly likely to have redundant functions with SET-24 are SET-9 and SET-26. These three SET domain-containing proteins are required for heritable RNAi, germline immortality at 25 °C, and interact with HCF-1[63,83]. While we identified HCF-1 as a SET-24 interactor (Fig. 5a, b, Supplementary Fig. 6b), IP-MS and Y2H analyses of the interactome of SET-24 did not detect SET-9 or SET-26. Likewise, SET-24 was not identified in the interactome of SET-9/26[63], suggesting that SET-24 does not interact directly with these other SET domain-containing proteins. Human HCF-1 interacts with MLL5 through its Kelch domain and HBM on MLL5[65]. In *C. elegans*, SET-9 and SET-26 contain the conserved HBM, which is thought to mediate their interaction with HCF-1[63] (Supplementary Fig. 2a). In contrast, SET-24 did not show a HBM of identical sequence. Instead, both its SET domain and the flanking sequences interact with the Kelch domain of HCF-1 (Fig. 5f-k and Supplementary Fig. 7). These findings suggest that SET-24 and SET-9/SET-26 are likely to engage the same domain of HCF-1 in distinct ways. At this point, we cannot exclude the alternative that SET-24 competes with SET-9/26 for HCF-1 binding.

The subcellular localization and expression pattern of these SET domain-containing proteins are also consistent with functional redundancy. Much like HCF-1, SET-9/26 and SET-24 are all expressed in germline nuclei[41,64] (Fig. 4a and Supplementary Figs. 6d-f), although HCF-1 and SET-26 are present in somatic nuclei as well. The expression pattern of SET-26 and HCF-1 is consistent with additional regulatory roles outside the germline. Indeed, SET-26 was shown to modulate the lifespan of *C. elegans* in conjunction with HCF-1 by binding to H3K4me3 in somatic cells[63]. In the germline, SET-24 displays a pattern of expression from the mitotic tip to the pachytene-diplotene boundary region (Fig. 4a), indicating that SET-24 might have a regulatory role up until the pachytene-diplotene transition and affect normal meiotic progression. In fact, this is a germline region where chromatin changes required for meiotic progression, chromosomal organization and recombination have been previously described[84–86]. Conversely, HCF-1 and SET-9/26 are expressed throughout the gonad[41,64] (Supplementary Fig. 6f). Therefore, in mitotic and early meiotic stages, SET-24 is co-expressed with SET-9/26 in germline nuclei, where these factors associate with HCF-1 and may have partial redundant regulatory roles. Both SET-24 and HCF-1 form nuclear puncta and colocalize with each other (Figs. 4b-d, 5d-e). Determining the precise nuclear localization of these puncta will be important for clarifying the functions of SET-24 and HCF-1 in future studies.

Small RNA-driven TEI consists of three steps: initiation, establishment, and maintenance[18]. Our RNAi inheritance assays show that SET-24 and HCF-1 act exclusively in the maintenance step. This is unusual, as most TEI factors, such as HRDE-1, WAGO-4, and ZNFX-1,

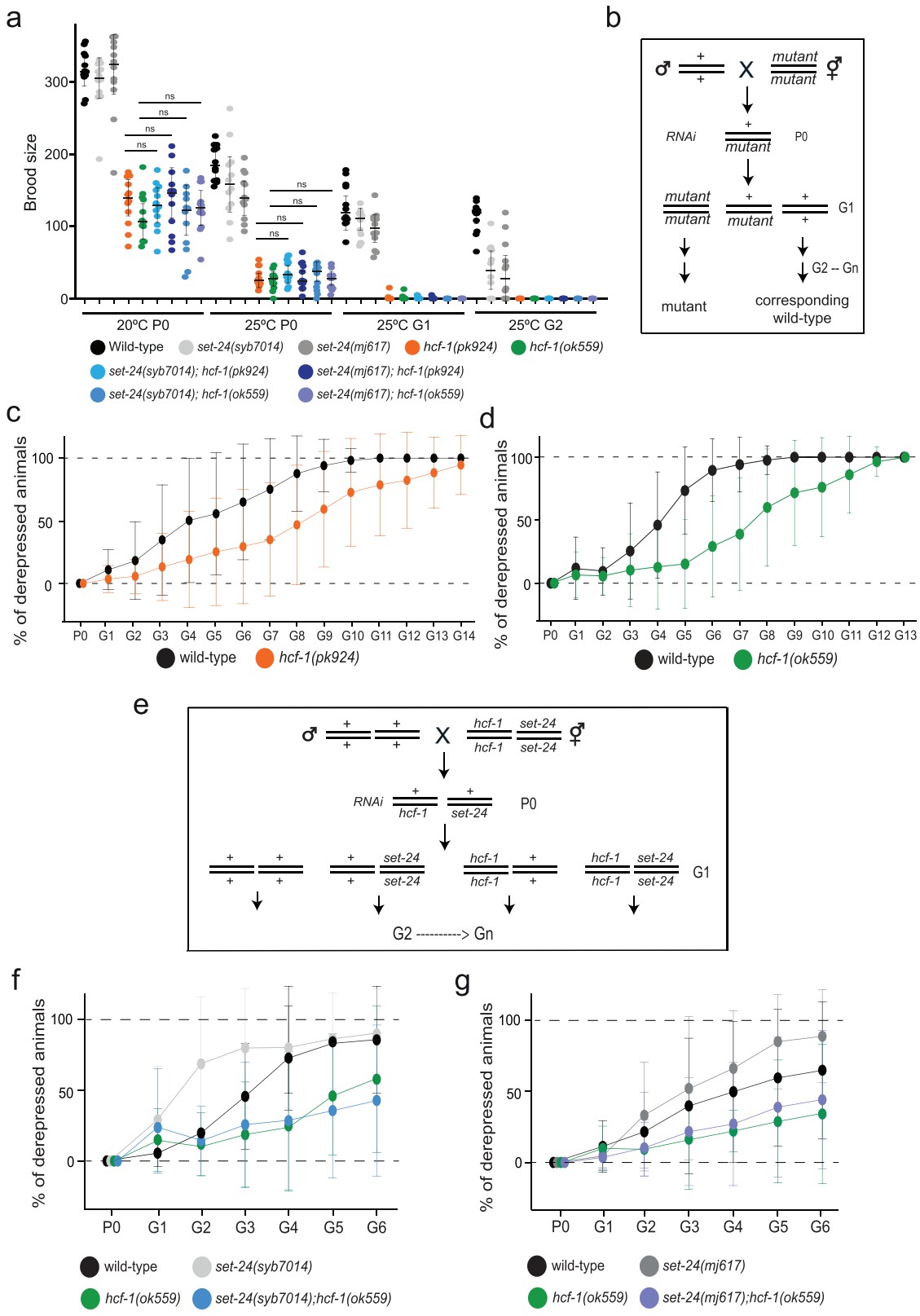

play roles in both the establishment and maintenance phases, whereas SET-25 and SET-32 are primarily involved in the establishment[27]. SET-24 and MET-2 are the only SET domain-containing proteins with an exclusive role in the maintenance stage[22]. Notably, SET-24 and HCF-1 contribute distinctly to the maintenance of heritable RNAi, as *set-24* and *hcf-1* mutants display opposite phenotypes: shortened and extended transgenerational RNAi-driven silencing, respectively

(Figs. 3b, 6b-d, Supplementary Fig. 8b). Phenotypically, *hcf-1; set-24* double mutants phenocopy *hcf-1* single mutants, suggesting that HCF-1 may function downstream of SET-24 in the maintenance of TEI (Fig. 6e-g and Supplementary Figs. 8c-f). This could also be explained by the myriad roles played by HCF-1 in the context of different protein complexes, compared to more specialized SET-24. Besides interaction with SET-24, HCF-1 interacts with the H3K4 methyltransferase SET-2

**Fig. 6 | HCF-1 inhibits the maintenance of heritable RNAi. a** Brood size of indicated strains at 20 °C and 25 °C over generations. Means and standard deviations are shown, $n = 12$ worms. Statistical significance was assessed using a two-sided unpaired *t*-test. ns, not significant. Source data and exact *p* values are provided as a Source Data file. **b** A scheme of RNAi inheritance assay for the below plots: these heterozygous individuals are fed with GFP RNAi bacteria. Silenced mutants and their corresponding wild-type siblings (all selected by genotyping PCR) are self-fertilize to produce the F1 and later generations, and are fed on HB101-seeded plates. **c, d** Quantification of the percentage of derepressed individuals at every generation. Means and standard deviations are shown; **c** *wild-type*, $n = 22$ animal lineages; *hcf-1(pk924)*, $n = 18$ animal lineages. **d** *wild-type*, $n = 22$ animal lineages; *hcf-1(ok559)*, $n = 22$ animal lineages. Source data for are provided as a Source Data file. **e** A scheme of RNAi inheritance assay for the plots below. Heterozygous individuals are fed with GFP RNAi bacteria. Silenced double mutants, single mutants, and wild-type siblings (all selected by genotyping PCR), are self-fertilized to produce the G1 and later generations, and are fed on HB101-seeded plates. **f** and **g** Quantification of the percentage of derepressed individuals at every generation. Means and standard deviations are shown; (**f**) $n = 8$ animal lineages; (**g**) $n = 9$ animal lineages. Source data for are provided as a Source Data file.

and the COMPASS complex, other chromatin remodelling complexes, and with the H3K4me3 readers SET-9 and SET-26[41,62–64,68–71,87,88]. These interactors have distinct effects on TEI and may explain the role of HCF-1 in inhibiting TEI. For example, while SET-9/26 influence TEI[83], SET-2 is not required for the process[89]. This role of HCF-1 in extending RNAi inheritance is not unprecedented, as similar roles have been reported for HERI-1, MET-2, and LOTR-1[19,22,90]. Whether HCF-1 is acting in concert with HERI-1, MET-2, or LOTR-1 remains to be determined. Further research is needed to better understand how SET-24 and HCF-1 regulate TEI.

Our data suggests an imbalance of H3K4me3 and 22G-RNA levels caused by *set-24* mutation (Figs. 3d and 7). We propose that this imbalance, together with our phenotypic data, reflects a role of SET-24 in the maintenance of epigenetic memory across generations. We found that H3K4me3 levels of the TSSs of 757 genes are regulated by SET-24, with an upregulation observed in *set-24* mutants that is not consistently accompanied by transcriptional activation of these genes. However, a larger subset of 220 genes with SET-24-dependent H3K4me3 regulation tend to have deregulated 22G-RNA levels. Genes with deregulated 22G-RNA levels are targets of silencing 22G-RNA pathways in wild-type (Fig. 7e and Supplementary Fig. 11d). The relatively small number of genes affected may be due to possible redundancy with other germline-expressed SET domain-containing proteins, as discussed above. We investigated the levels of H3K4me3 given the association of SET-24 with HCF-1, which is a cofactor of H3K4me3-directing COMPASS complex. However, the lack of transcriptional upregulation of SET-24-dependent H3K4me3-enriched genes may be due to other repressive chromatin modifications that were not profiled in our study, such as H3K9me3. As H3K9me3 is a mark associated with the WAGO and mutator 22G-RNA silencing pathways targeting SET-24-dependent genes, we postulate that the interplay between H3K4me3 and H3K9me3 is disrupted in *set-24* mutants and affects the maintenance of the chromatin environment adequate for the maintenance of gene silencing across generations. Lack of sustained 22G-RNA biogenesis may in turn contribute to further destabilization of the chromatin environment at these loci. Further genetic and biochemical dissection of these processes is required to understand how SET-24 and other SET domain-containing proteins define the chromatin landscape and affect 22G-RNA biogenesis in the germline.

The *set-24* allele was originally identified in wild *C. elegans* isolates with a mortal germline[35]. How could mutations causing a Mrt phenotype be maintained in wild populations? Perhaps the answer lies in the "escapees" identified in the fertility assays across generations (Fig. 2). The Mrt phenotype is reversed in particular animal lineages that presumably develop a compensatory response, mitigating the impact of these mutant alleles on the transgenerational fertility of wild populations. Interestingly, escape from the Mrt phenotype also occurs occasionally in *set-9* and *set-26* single mutant lines[83]. The "escapee" phenotype could reflect a compensatory response from redundant germline-expressed SET domain-containing factors. Reversibility of the Mrt phenotype is not unprecedented, for example upon exposure to specific stimuli, such as temperature shifts and different bacterial diets[34,35]. What possible advantage, if any, could such a mutant allele

confer in the wild? A provocative hypothesis consists in the establishment of secondary chromatin-state and small RNA-based epimutations, some of which may be advantageous. In line with this hypothesis, prominent 22G-RNA-based epimutations in WAGO targets have been documented[91]. Therefore, natural genetic variation may disrupt epigenetic processes, like in the wild-isolated strain defective for *set-24*, leading to secondary epimutations on the chromatin state and small RNA regulation. These epimutations could help wild worm populations deal with and adapt to environmental fluctuations, supporting the essential roles played by small RNA pathways in pathogen and environmental stress responses[92–100].

## Methods

### Evolutionary and structural analysis of SET and SPK domains

As the hidden Markov model (HMM) for SET domains within Pfam-A.hmm (version 3.1.b2, Feb 2015)[101] was not sensitive enough to retrieve the SET domain of SET-24 of *C. elegans*, we first constructed an alternative HMM. To do so, we first downloaded the protein sequences of the InterPro[102] SET domain (IPR046341), and the metadata with the associated domain coordinates, filtering for all human SET domains, as well as the catalytically inactive SET proteins of *Drosophila melanogaster* (UpSET) and *Saccharomyces cerevisiae* Set3. Then, we used the domain coordinates to trim all these protein sequences, keeping only the sequences of the SET domains, which were used as input for multiple sequence alignment with MAFFT v7.475, using option --auto[29,103]. Then, the HMM profile was built with this alignment using hmmbuild of the HMMer package (v3.3, hmmer.org). The HMM profile was used to search the entire *C. elegans* proteome (Wormbase ParaSite, version WBPS16)[104] for SET domains with hmmsearch of the HMMer package (version 3.3, hmmer.org), with option --nobias. The *C. elegans* SET domains were aligned with MAFFT v7.475[103], using option --auto (model chosen L-INS-i). The resulting alignment was input to infer a maximum likelihood phylogenetic tree with IQ-TREE v2.1.2[105] with 1000 bootstraps (option -B 1000)[106]. LG + G4 was the best fit model. We created an additional phylogenetic tree using the above-mentioned human SET domains, plus the SET domain of *C. elegans* SET-24. Alignments were conducted with MAFFT (L-INS-i was the chosen model), and a tree was constructed with IQ-TREE (LG + G4 was the best fit model) as above.

We obtained the sequences of all proteins with an SPK domain (IPR006570), and the coordinates of the SPK domains from InterPro[102]. Sequences were trimmed according to the SPK coordinates to leave only the SPK sequence. The SPK domains were subsequently aligned with MAFFT v7.475[103], using option --auto (model chosen L-INS-i), and a maximum likelihood tree was constructed with IQ-TREE v2.1.2[105,106] with option -B 1000, and with LG + G4 as the best fit model.

The structure of SET-24 was predicted with AlphaFold3[107], and the regions corresponding to the annotated SPK domain coordinates were extracted and used as input in Foldseek[108]. The MYB domains of TEBP-1 and TEBP-2 were amongst the top hits identified by Foldseek. We used AlphaFold3[107] to predict the structures of TEBP-1 and TEBP-2, and extracted the structures corresponding to their MYB domains, according to previously defined domain annotations[51] and predicted

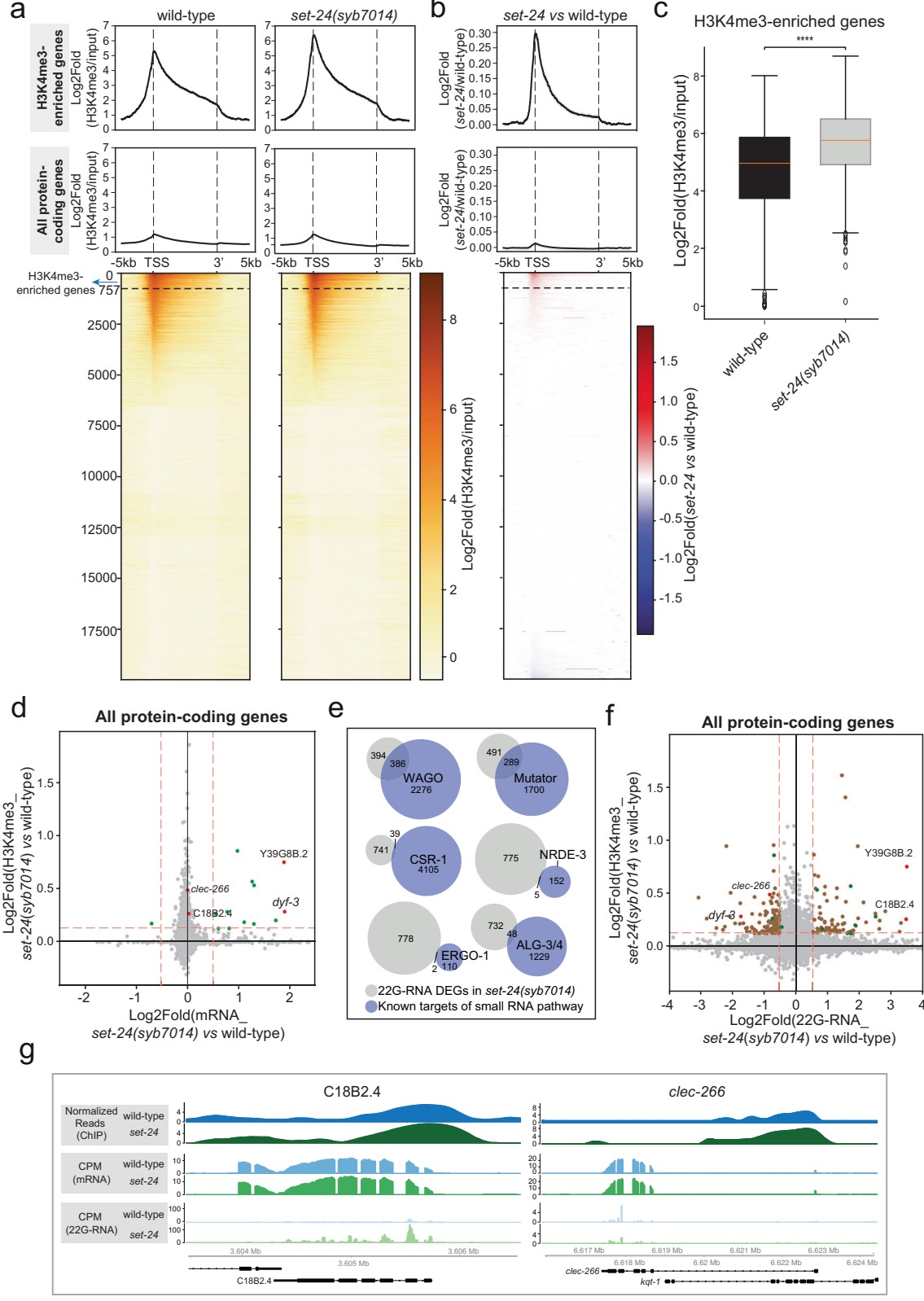

secondary structure elements. ChimeraX[109] was used to visualize predicted models and perform structural alignments.

## Strains

The Bristol strain N2 was used as the standard wild-type strain. All strains were grown at 20 °C unless otherwise specified. The strains used in this study are listed in Supplementary Data 8.

## Construction of transgenic strains

Wild-type YA animals (N2 strain) were injected with a mixture of target gene HR repair template (IDT oligos) (1 mg/ml), target gene CRISPR crRNA (Dharmacon) (8 mg/ml) and His-Cas9 (in-house bacterial purification) (5 mg/ml) dissolved in injection buffer (10 mM KCL; 10 mM Tris–HCl at pH: 8.0). F1 animals were singled, allowed to produce homozygous F2 progeny by selfing and genotyped. All final strains

**Fig. 7 | SET-24 modulates H3K4me3 accumulation and small RNA biogenesis.**
**a**, **b** Metagene plots and heatmaps showing H3K4me3 enrichment normalized to
input for wild-type samples and *set-24* mutants across H3K4me3-enriched genes
and all protein-coding genes (**a**) and the log₂-normalized fold-change between *set-
24(syb7014)* mutants and wild-type samples (**b**). A subset of H3K4me3-enriched
genes that contribute to 80% of the total increased enrichment in *set-24(syb7014)*
mutants is labelled in the heatmaps. H3K4me3 ChIP-seq of *set-24(syb7014)* and wild-
type. $n = 2$ independent replicates. **c** Comparison of the average H3K4me3
enrichment around the TSS ( ± 500 bp) for wild-type samples and *set-24(syb7014)*
mutants across H3K4me3 enriched genes. Statistical significance was assessed
using a two-sided unpaired *t*-test. Stars indicate *p* value, ****p* value < 0.0001. The *p*
value for the comparison between wild-type and *set-24(syb7014)* is 9.27e⁻⁰⁷. $n = 757$
H3K4me3 enriched genes. Box whiskers represent maxima and minima, center lines
show median, box bounds show the quartiles, and dots show outliers. Source data
are provided as a Source Data file. **d** Scatter plot correlating the increased H3K4me3
enrichment around the TSS and mRNA levels of protein-coding genes. These genes
were selected from those with H3K4me3 increase that also exhibited a log₂-
normalized increase of ≥0.5 in mRNA levels. mRNA-seq of *set-24(syb7014)* and wild-
type, $n = 4$ independent replicates. **e** Venn diagrams showing overlaps between lists
of genes with deregulated 22G-RNA levels in *set-24(syb7014)* mutants (fold change >
2 and 5% FDR) and genes targeted by specific small RNA pathways. Small RNA-seq of
*set-24(syb7014)* and wild-type, $n = 4$ independent replicates. **f** Scatter plot corre-
lating the increased H3K4me3 enrichment around the TSS and 22G-RNA levels
across all protein-coding genes. These genes were selected from those with
H3K4me3 increase that also exhibited a log₂-normalized change of ≥0.5 in 22G-RNA
levels. **g** H3K4me3 enrichment, mRNA and 22G-RNA levels of the selected genes
(C18B2.4 and *clec-266*) in wild-type and *set-24(syb7014)* mutant strains.

were checked by Sanger sequencing and outcrossed twice. The *set-24
(syb7014) and the set-24::3xflag (syb4492)* strains were made by Suny-
Biotech. The *set-24 (mj617) and set-24::gfp(mj616)* strains were made by
standard CRISPR/Cas9 methods[110]. Fot the *set-24::gfp (mj616)* strain, the
homologous repair template containing the GFP sequence generated
by PCR from the AP625 plasmid (Addgene), and a guide RNA, TGAT-
CATTTCGATGACAACG, were used. For the *set-24 (mj617)* strain, two
guide RNAs, TGATCATTTCGATGACAACG and ACAGCCGGTGAGAA
TGTGTT, and the repair temple, AAAAAAAAACAAAGGAGAGATGCA
TTAACTTGTAAGAAAATAATATTATCATTGAACATCCGGCGAATTTTG
GAGAAAACTGGTGGTTTCGTTGAAATCCATGATTGTTCTTGTGAAATT
ATACACTGAAAATAAATATTTATATGTATTACTTATTTTTAAATATTTA
GT, were used. For generating the *set-24(mj673)::gfp, set-24(mj674)::gfp,
set-24(mj675)::3xflag, set-24(mj676)::3xflag*, and *set-24(mj677)::3xflag*
strains, pDD162, an injection marker, three plasmids expressing
sgRNAs targeting the *set-24* sequence, and/or a repair template were
co-injected into the germline of *set-24::gfp (mj616)* or *set-24::3xflag
(syb7014)* animals. For the *Pmex-5::hcf-1::mKate2 (mjIS606)* strain,
pDD162, an injection marker, three sgRNA-expressing plasmids
(GAAATCGCCGACTTGCGAGG, GCAATGACTAACCGATTTTC, and
TTCGGGATAATTGAGATGAG) targeting the ttTi4348 site on Chro-
mosome I, and a plasmid containing *Pmex-5::hcf-1::mKate2::hcf-1 3'UTR*
with repair templates were co-injected into the *C. elegans* germline.
Mutants or transgenes were identified by PCR screening of F1 worms
carrying the injection marker.

### Germline mortality assay
Worms were maintained at 20 °C prior to the start of the experiment.
At least 10 L4-stage individuals per genotype, designated as the P0
(parental) generation, were transferred to HB101-seeded plates at
25 °C and allowed to produce offspring. At each generation, one L3
larva per replicate per genotype was transferred to a new plate to
produce the next generation. The number of sterile animals at each
generation was recorded.

### Brood size assay
L3 worms were individually placed onto fresh NGM plates. Offspring
counting began once the progeny reached the L4 stage, and counted
offspring were removed using a vacuum pump. The number of pro-
geny was counted in each generation over the course of three
consecutive days.

### Transgenerational memory inheritance
One L4 larva per genotype was plated on either GFP RNAi-expressing
bacteria or empty vector L4440 bacteria. G1 animals were examined
under a fluorescence microscope, and one silenced animal per repli-
cate per genotype was transferred onto plates seeded with standard
HB101 bacteria. At each generation, a single silenced animal was iso-
lated from each plate to produce the next generation, while the
remaining adult progeny were analysed under a fluorescence

microscope. At least 20 animals per replicate per genotype were
counted at each generation unless otherwise specified. For Fig. 3b,
germline nuclear GFP brightness was scored by visual inspection as 1
for "bright" and 0 for "dark", with dim GFP expression also classified as
0. For Fig. 6c, d, f, and g, Supplementary Figs. 8a, c-f), nuclear GFP
brightness was assessed visually, with "bright" scored as 1, "dark"
scored as 0, and "dim" scored as 0.5. The percentage of derepressed
worms was calculated as the sum of the scores divided by the total
number of worms examined. Representative images were captured
using a Leica SP8 fluorescence microscope at 40X magnification.

### RNA extraction and real-time quantitative PCR
For wild-type and *set-24* worms, we picked one repressed worm at each
generation (when available) and extracted RNA from its progeny, after
setting aside one repressed worm to propagate the next generation.
For the *hrde-1* mutant, we picked derepressed worms starting from G1.
Total RNA was extracted using TRIzol reagent (Ambion, Life Tech-
nologies) and treated with Turbo DNase Kit (Invitrogen) according to
the manufacturer's instructions. 500 ng of total RNAs per sample were
reverse-transcribed with random hexamers (Invitrogen) at 50 °C for 1 h
using Superscript III (Invitrogen). Reactions lacking reverse tran-
scriptase were systematically run in parallel as negative controls. Real-
time quantitative PCR was performed on 1 ul of diluted (1/5) RT reac-
tions using SYBR Green kit (Life Technologies) on a OneStepPlus
thermocycler (Thermo Fisher). All samples were run in duplicates and
expression levels normalized to the reference gene *cdc-42* according to
the ΔΔCt method[111]. qPCR primers used in this work are listed in Sup-
plementary Data 9.

### 22G-RNA real-time quantitative PCR
50 ng of total RNA were reverse-transcribed with a TaqMan Small RNA
Assay Kit (Thermo Fisher) containing a gene-specific RT primer and a
TaqMan MicroRNA Reverse Transcription Kit (Thermo Fisher),
according to the manufacturer's instructions. Real-time quantitative
PCR was performed on 1 ul of diluted (1/5) RT reactions using custom-
made TaqMan probe (GUGUCCAAGAAUGUUUCCAUCU), TaqMan
Universal Master Mix No AmpErase UNG (Life Technologies) on a
OneStepPlus thermocycler (Thermo Fisher). All samples were run in
triplicates. Expression levels were normalized to the reference gene
U18 (#001764, TaqMan) according to the ΔΔCt method[111].

### Whole-mount DAPI staining
At every generation, synchronized adult worms were collected in M9,
fixed in 70% ethanol for 1 h, centrifuged and washed twice with 0.1%
Tween-20/PBS (PBST) and strained in 100 mg/mL DAPI/PBST for
30 minutes at room temperature on a rotating wheel. Worms were
then washed twice with PBST, pipetted onto a microscope slide and
mounted in Vectashield. Worms were classified in categories and
manually counted (at least 50 animals per genotype per generation
were counted) under a fluorescence microscope. Single-plan

representative images were taken on a SP8 confocal fluorescence microscope (Leica) at 63X magnification.

## Immunofluorescence staining

Worms were picked on glass slides and manually dissected to extrude the germlines. Germlines were subsequently freeze-cracked on dry ice and fixed in cold 100% methanol for 20 min. Fixed slides were then washed in 0.05% Tween/PBS and incubated with diluted (1:1,000) anti-GFP (Abcam #ab290) and diluted (1:1,000) anti-PGL-3 (a gift by Susan Strome) antibodies overnight. The following day, slides were washed in 0.05% Tween/PBS and incubated with diluted secondary antibodies for 1 h at room temperature and mounted in Vectashield with DAPI. Representative images were taken on a Leica SP8 fluorescence microscope with a 63X oil objective and 4X digital magnification. Single-plan images are shown.

## Imaging

A 4% agarose pad was prepared on a glass slide. Gravid adult animals were picked into a droplet of M9 buffer to remove residual *E. coli* and then transferred to another droplet of M9 on the agarose pad supplemented with 5 mM tetramisole hydrochloride for immobilization. Images for Figs. 2c, 3b, and 4a, Supplementary Figs. 5g, 6d, and 6e were acquired using a Leica SP8 fluorescence microscope. Z-stack images for Figs. 4b, 4c, 4f and 5d, Supplementary Figs. 5f, 6f, and 6h were obtained using a Nikon Ti2-E inverted microscope (NIS-Elements AR v5.42.06) equipped with Plan Apo VC 20×/0.75 air and 60×/1.2 water objectives. Images were processed in Fiji v2.16 by adjusting brightness and contrast.

Co-localization was quantified in Fiji v2.16. Germline nuclear regions displaying SET-24::GFP signals were extracted from confocal images derived from more than four germlines, with 20 images analysed to calculate co-localization coefficients. Coefficients were determined from multiple confocal planes of the same nucleus. Pearson's correlation coefficient was used for co-localization analysis, and results are reported in the main text.

## SET-24::3XFLAG immunoprecipitation and mass spectrometry

Procedure was conducted as previously described[19,50]. Wild-type N2 and SET-24::3xFLAG animals were grown at 20°C in HB101 high-density plates synchronized by bleaching and overnight hatching of L1s in M9 buffer. L1s were plated and grown at 20°C for 51–55 h, until the YA stage. At this stage, worms were washed off plates, washed 3-4 times in M9 buffer, washed one last time with deionised water, and snap-frozen on dry ice. To prepare extracts, worm samples were thawed and mixed 1:1 with 2x Lysis Buffer (50 mM Tris/HCl pH 7.5, 300 mM NaCl, 3 mM MgCl2, 2 mM DTT, 0.2 % Triton X-100, and complete EDTA-free Mini protease inhibitors, Roche #11836170001). Lysis was subsequently performed by sonication in a Bioruptor Plus (Diagenode, on high level, 10 cycles of 30 seconds on and 30 seconds off). After sonication, the samples were centrifuged at 21,000 x *g* for 10 min to pellet cell debris, and the supernatant was transferred to a fresh tube. Protein concentrations were determined with Bradford Protein Assay (according to manufacturer's instructions, Bio-Rad, #5000006). IPs were prepared in quadruplicates for each strain used. 30 µl of Dynabeads Protein G (Invitrogen, #10003D) were used per IP and washed three times with 1 ml Wash Buffer (25 mM Tris/HCl pH 7.5, 300 mM NaCl, 1.5 mM MgCl2, 1 mM DTT, and complete EDTA-free Mini protease inhibitors, Roche #11836170001). The beads were resuspended in Wash Buffer and combined with to 2 mg of complete protein extract, for a total volume of 500 µl. Finally, 2 µg of anti-FLAG antibody (Sigma-Aldrich, #F1804) were added, and the samples were incubated for 3h30m, rotating at 4 °C. After the incubation, the samples were washed five times with 1 ml Wash Buffer, followed by bead resuspension in 1x LDS/DTT, and boiling at 95 °C for 10 min.

IP samples were boiled at 70 °C for 10 minutes and separated on a 4–12% gradient Bis-Tris gel (Thermo Fisher Scientific, #NP0321) in 1x MOPS (Thermo Fisher Scientific, #NP0001) at 180 V for 10 minutes. Then, samples were processed separately, first by in-gel digestion, followed by desalting with a C18 StageTip[112,113]. Afterwards, the digested peptides were separated on a heated 50-cm reverse-phase capillary (75 µm inner diameter) packed with Reprosil C18 material (Dr. Maisch GmbH). Peptides were eluted along a 90 min gradient from 6 to 40% Buffer B (see StageTip purification) with the EASY-nLC 1200 system (Thermo Fisher Scientific). Measurement was done on an Orbitrap Exploris 480 mass spectrometer (Thermo Fisher Scientific) operated with a Top15 data-dependent MS/MS acquisition method per full scan. Four replicates were performed.

All raw files were processed with MaxQuant[114](version 1.6.5.0) and peptides were matched to the *C. elegans* Wormbase protein database (version WS269) including *E. coli* sequences (ASM1798v1). Raw data and detailed MaxQuant settings can be retrieved from the parameter files uploaded to the ProteomeXchange Consortium via the PRIDE repository, accession number PXD057349. Data analysis was completed in R. Statistical significance was assessed using a two-sided unpaired *t*-test.

## Yeast-two hybrid screening

Yeast two-hybrid (Y2H) "prey" library construction, screening, and classification were performed by HYBRiGENiCS SERVICES. For "prey" library construction, mRNAs were purified from ~1 mg of total RNA extracted from mixed-stage *C. elegans*. The resulting cDNAs were ligated into the pP6 vector containing the LEU3 selectable marker. For "bait" construction, the full-length *set-24* open reading frame was cloned into the Gateway pBD-GAL4 vector containing the TRP1 selectable marker. "Bait" and "prey" constructs were transformed into the yeast strain CG1945 and brought together by mating. Positive colonies were selected on dropout media lacking tryptophan, leucine, and histidine, and supplemented with 0.5 mM 3-amino-1,2,4-triazole to suppress weak self-activation by SET-24. Recovered "prey" clones were amplified by PCR, sequenced, annotated, and assigned a Predicted Biological Score (PBS) ranging from very high confidence (A) to lower confidence (D).

## Protein purification and pull down

The coding sequences of HCF-1, SET-24, and the truncations were amplified using cDNA template reversed transcribed from *C. elegans* total RNA and cloned into the pGEX6p (GST tag) and pMAL (MBP tag) plasmids. The plasmids were transformed into *E. coli* Rosetta (DE3) cells (Merck, #71403) for expression. The proteins were induced with 0.1 mM IPTG at 16°C for 16-20 h and collected by centrifugation. Rosetta (DE3) cells were resuspended in the purification buffer (50 mM Tris-HCl PH 7.5, 150 mM NaCl, 1 mM DTT, Protease inhibitor, and 1% Triton-X 100) and lysed by sonication. The cell lysate was then centrifuged at 20, 000 rpm for 20 minutes at 4°C. The supernatant was subjected to affinity purification using GST beads (Cytiva, #17527901) or MBP beads (NEB, #E8022L). Then, the beads were washed three times with washing buffer (50 mM Tris-HCl PH 7.5 and 150 mM NaCl). For the GST-tagged proteins, resuspend the beads with washing buffer + 10% glycerol. For the MBP tagged proteins, the bound proteins were eluted using MBP elution buffer (500 mM maltose, 50 mM Tris-HCl PH 7.5, 500 mM NaCl, and 1 mM DTT), and then dialysed with washing buffer + 10% glycerol. The concentrations of the eluted proteins were measured by Bradford assay (Thermo fisher, #A55866). The beads bound GST tagged proteins and the MBP tagged proteins were aliquoted, flash-frozen in liquid nitrogen, and stored at -80°C.

The GST-tagged proteins were immobilized on GST beads and incubated with MBP-tagged proteins in 500 µl binding buffer (20 mM Tris-HCl PH 7.5, 150 mM KCl, 2 mM MgCl2, 1 mM DTT, 0.01% Triton-X 100, and 5% Glycerol) on a rotator at room temperature for 1 hour.

Then, the beads were washed with 1 mL binding buffer for 10 times and boiled with 5x loading buffer. The samples were separated with SDS-PAGE and the proteins were detected with Western blotting.

## Western blotting

Synchronized YA stage worms either growing at 20 °C or three generations at 25 °C were harvested and washed three times with M9 buffer before being frozen at -80 °C. Worm proteins were extracted by heating the samples at 95 °C for 10 minutes in 1× protein dye (62.5 mM Tris pH 6.8, 10% glycerol, 2% SDS, 5% β-mercaptoethanol, 0.2% bromophenol blue). Samples were then spun at high speed for 1 minute to remove insoluble components, and the supernatant was quickly transferred to a new tube on ice. The samples were either immediately loaded onto a gel or stored as at −80 °C.

Proteins were separated by SDS-PAGE on gradient gels (10% separation gel, 5% spacer gel) and transferred onto a Hybond-ECL membrane. After washing with 1× TBST buffer (20 mM Tris pH 7.4, 150 mM NaCl, and 0.1% Tween 20) and blocking with 5% milk-TBST or BSA-TBST, the membrane was incubated overnight at 4 °C with primary antibodies (listed below). The next day, the membrane was washed three times for 10 minutes each with 1× TBST, followed by incubation with secondary antibodies at room temperature for 2 h. After three additional 10-minute washes with 1× TBST, the signal was visualized.

For gel reblotting, membranes were stripped with strip buffer (62.5 mM Tris pH 6.8, 10% glycerol, 2% SDS, 0.7 % β-mercaptoethanol) for 1 h and then washed three times with 1× TBST. The stripped membranes were subsequently blocked with 5% milk–TBST or BSA–TBST and incubated with primary and secondary antibodies as described above.

The primary antibodies used were β-actin (Beyotime, AF5003, 1:1000), H3 (Abcam, ab1791, 1:1000), H3K4me2 (Abcam, ab7766, 1:1000), H3K4me3 (Abcam, ab8580, 1:1000), H3K9me1 (Abcam, ab9045, 1:1000), H3K9me2 (Abcam, ab1220, 1:1000), H3K9me3 (Millipore, 07-523, 1:2000), H3K23me2 (Active Motif, 39653, 1:1000), H3K23me3 (Active Motif, 61499, 1:1000), H3K27me3 (Millipore, 07-449, 1:1000), H3K36me3 (Abcam, ab9050, 1:1000), H3K27Ac (Abcam, ab4729, 1:2000), H3PanAc (Abcam, ab47915, 1:1000), H4PanAc (Abcam, ab177790, 1:1000), H4 (Abcam, ab10158, 1:1000), Tubulin (Abcam, ab6160, 1:3000), FLAG (Sigma, F1804, 1:1000), GFP (Abcam, ab290, 1:1000), GST (Abcam, ab9085, 1:1000), MBP (Abcam, ab49923, 1:1000; Abcam, ab9084, 1:5000). The secondary antibodies used were goat anti-mouse (Beyotime, A0216, 1:5000; Abcam, ab6789, 1:5000), goat anti-rabbit (Abcam, ab205718, 1:20000; Abcam, ab6721, 1:20000), and goat anti-rat (Abcam, ab6734, 1:5000).

## In vitro methyltransferase assays

This assay was performed using the non-radioactive MTase-Glo™ Methyltransferase Assay Kit (Promega, V7601). Briefly, purified proteins were incubated with histones (Roche #10223565001) (or non-histone controls) and SAM in reaction buffer (25 mM Tris-HCl pH 8.3, 50 mM KCl, 10 mM MgCl₂, 1 mM DTT, and 0.1 µg/µl BSA). The reactions were then incubated for 1 h at 30 °C and terminated by addition of trifluoroacetic acid (TFA) to a final concentration of 0.125%. MTase-Glo reagent and MTase-Glo detection solution were subsequently added, and luminescence was measured following manufacture instructions.

## ChIP and ChIP-seq analysis

ChIP protocol was based on a previously published protocol[115]. Synchronized worms at the YA stage were collected in M9 buffer and rapidly frozen in liquid nitrogen to create worm pellets. The worm pellets were transferred to a metallic grinder that had been pre-cooled in liquid nitrogen for 5 minutes. The worms were ground until broken into small pieces while keeping the nuclei intact. The resulting powder was transferred into a cold 50 mL Falcon tube. The crosslinking was performed on a 40 ml PBS solution containing 1% formaldehyde by shaking at room temperature for 8 minutes. Then, 4.6 mL of 1.25 M glycine was added to quench the reaction, and the mixture was gently shaken for another 8 minutes at room temperature. The sample was washed twice in PBS with protease inhibitor (PI, cOmplete Tablets, Mini EDTA-free, *EASYpack*, REF #04693159001) and once in FA buffer (50 mM Hepes/KOH pH 7.5, 1 mM EDTA, 1% Triton X-100, 0.1% sodium deoxycholate, and 150 mM NaCl) with PI. The worms were sonicated for 25 cycles of 30 seconds on and 30 seconds off. A 30 µL aliquot was crosslinked to be used as input. The remaining mixture was centrifuged at 4 °C for 15 minutes, and the supernatant was transferred into a new tube and frozen at -80ºC. The immunoprecipitation was done by adding antibody (2 ug of H3K4me3 antibody, Active Motif, #39159 or H3 antibody, Abcam, #ab1791), then rotated overnight at 4 °C. Next, 40 µL of beads (DynabeadsTM Protein A, Invitrogen, REF #10004D) were taken and washed twice with 1 mL of FA + PI. The beads were resuspended in 1 mL of FA + PI + 1% BSA + 10 µL of tRNA and rotated overnight at 4 °C. The beads were washed twice with FA + PI and then transferred into the extract/antibody solution, followed by rotation at 4 °C for 2 h. The beads were then washed twice in FA + PI, once in FA with 500 mM NaCl, once in FA with 1 M NaCl, and twice in TEL buffer (0.25 M LiCl, 1% IGEPAL, 1% sodium deoxycholate, 1 mM EDTA, 10 mM Tris-HCl pH 8). The beads were eluted in 60 µL of ChIP elute buffer and incubated at 65 °C for 15 minutes. The elution was transferred to a new tube as the IP. For decrosslinking, 2 µL of RNase (Roche, 1119915001) was added, and the mixture was incubated at 37 °C for 1.5 h. Finally, 1.5 µL of proteinase K (20 mg/mL, NEB, P8107S) was added, and the mixture was incubated overnight at 65 °C for decrosslinking. The DNA was purified from the solution using a PCR purification kit (Invitrogen, K31002).

DNA libraries were prepared and sequenced by Novogene. The purified DNA samples were treated with End Repair Mix (Novogene) and incubated at room temperature for 30 minutes. They were then purified using a PCR purification kit (Qiagen). The DNA was subsequently incubated with A-tailing mix at 37 °C for 30 minutes. Next, the 3′-end adenylated DNA was ligated with the adapter in the ligation mix at 20 °C for 15 minutes. The adapter-ligated DNA underwent several rounds of PCR amplification and was purified using a 2% agarose gel to recover the target fragments. The average fragment length was assessed using an Agilent 2100 Bioanalyzer (Agilent DNA 1000 Reagents) and quantified by qPCR (TaqMan probe). The libraries were further amplified on a cBot system to generate clusters on the flow cell and sequenced on an Illumina Novaseq X plus system (paired-end 50 base read length).

Adapter sequences were removed using Cutadapt 1.18[116] in the pair-ended read mode. Reads were then mapped to the *C. elegans* genome (WBcel235) using the Burrows-Wheeler Aligner with the MEM algorithm (BWA 0.7.17-r1188)[117]. Mapped reads were indexed and sorted using Samtools 1.10[118]. Bam files were filtered with samtools to remove non-unique mappers, secondary alignments and low-quality pairs (MAPQ < 10). Duplicate reads were removed using Picard MarkDuplicates 3.1.0-3[119] with the --REMOVE_DUPLICATES option. Peak calling was performed using Macs3 CallPeak (v3.0.0b1)[120] with no cutoff (-q 1) and the options --extsize 200 and --nomodel. Differential binding analysis was conducted using DiffBind (v3.12.0)[121] in R (v4.3.3) over a 200 bp sliding window with 10 bp shift across the entire genome. Counts were obtained without merging overlapping peaks and without computing summits. Normalization was performed using the DESeq2 method[122], accounting for library size, and differential analysis followed DiffBind's DESeq2 implementation.

BedGraph files displaying enrichment normalized to input for wild-type and *set-24* mutants, along with fold-change between *set-24* and wild-type enrichment, were generated using a custom Python script with a bin size of 10 bp. Metagene analysis was performed over all protein-coding genes identified in Ensembl's annotation (release

112) for WBcel235. The average H3K4me3 enrichment at TSS was calculated within a ± 500 bp window. Genes contributing to 80% of the total enrichment were selected for further analysis and classified as H3K4me3-enriched genes. All figures were generated using custom Python scripts. Analyses were performed using Python (v3.8.10) with Pandas (v2.0.1)[123] for data management, NumPy (v1.23.5)[124] for calculations, SciPy (v1.3.3)[125] for statistical tests, and Matplotlib (v3.4.3)[126] for figure generation.

## mRNA-seq and sequencing analysis

mRNAs were purified from total RNA using PolyT oligo-attached beads and converted to cDNAs for library preparation. cDNA libraries were sequenced using a paired-end 150 bp sequencing strategy on an Illumina Novaseq X plus system. Raw reads were assessed for quality using FastQC, Picard Tools, and Samtools along with trimming of poor-quality reads using Trimmomatic (v0.39; paramters: SLIDINGWINDOW:4:20, MINLEN:36, ILLUMINACLIP:TruSeq2-PE.fa:2:30:10)[118,127,128]. Clean reads were processed to find transcript abundance counts with Salmon (v1.10.2; paramters: --gcBias, --seqBias)[129]. We used DESeq2 (v1.46.0; padj <0.01) to identify differentially expressed genes in set-24 mutants compared to the WT[122]. Statistical significance was assessed using a two-sided unpaired t-test.

## small RNA-seq and sequencing analysis

Total RNAs were treated with RppH (NEB #M0356S)[130], and small RNA libraries were prepared with a small RNA-seq Kit v4 with UDIs (Nextflex #NOVA-5132-31). Libraries were sequenced using a single-end 50 bp sequencing strategy on a Novaseq6000 platform. Adapters and reads shorter than 18 nucleotides were removed using CutAdapt v1.15[131], with options -a TGGAATTCTCGGGTGCCAAGG --minimum-length 18. The quality of raw and trimmed reads was assessed with fastQC v0.11.9[127]. Subsequently, 22G-RNAs, defined as all reads between 21 and 23 nucleotides long starting with a G, were isolated. This filtering was conducted using a combination of CutAdapt v1.15[131], with options --minimum-length 21 --maximum-length 23, and zcat/awk utilities. 22G-RNAs were mapped to the C. elegans genome (WBcel235) using STAR v2.7.3a[132], with options --readFilesCommand zcat --outMultimapperOrder Random --outFilterMultimapNmax 100 --outFilterMismatchNmax 0 --alignIntronMax 1 --outSAMtype BAM SortedByCoordinate --outFilterType BySJout --winAnchorMultimapNmax 100 --alignEndsType EndToEnd --scoreDelOpen -10000 --scoreInsOpen -10000 --outSAMmultNmax 1. Subsequent quantification of counts and differential expression analysis were conducted as previously reported[133]. In short, featureCounts v2.0.0[134] was used to calculate counts mapping to genes (-t exon), using the BAM files produced in the previous STAR alignment step as input. The resulting tables of counts were imported into R, and DESeq2[122] and custom scripts (available online)[135] were used to obtain normalized counts and conduct statistical tests. The following R packages were used: tidyverse[136], lattice[137], eulerr[138], genefilter[139], reshape2[140], ashr[141], GenomicFeatures[142]. Genes with differentially expressed 22G-RNA levels in set-24(mj617) mutants, defined by fold change > 2 and 5% FDR, were overlapped with known target genes of specific small RNA populations using BioVenn[143]. Statistical significance was assessed using a two-sided unpaired t-test. We used previously described lists of known target genes[144–148].

## Generation of genome tracks

bedGraph files with ChIP-sequencing read counts normalized to library size and to input were converted to bigWig format using bedGraphToBigWig v2.8[149]. Small RNA and mRNA bigwig files and genome tracks were created as previously described[133]. In short, bigWigs were generated with bamCoverage v3.5.1[150], using options --normalizeUsing CPM --binSize 5 (for small RNA) or –binSize 10 (for mRNA) and the BAM files with 22G-RNAs/mRNAs mapped to the C. elegans genome. All the replicates of the same strain, either wild-type N2, set-24(syb7014), or set-24(mj617) mutants, were combined using WiggleTools mean[151] and wigToBigWig v4[149]. Genome tracks were plotted with custom scripts[135], on an R framework (R Core Team 2021), using the Gviz[152] and GenomicFeatures[142] packages.

## Statistics and Reproducibility

All experiments were conducted with independent C. elegans animals for the indicated n times. Statistical analysis was performed as indicated in the figure legends.

## Reporting summary

Further information on research design is available in the Nature Portfolio Reporting Summary linked to this article.

## Data availability

Sequencing data have been deposited in NCBI's Gene Expression Omnibus and are accessible through GEO Series accession number GSE291568. Proteomics data are available at the ProteomeXchange Consortium via the PRIDE partner repository with the identifier PXD057349. Source data are provided with this paper.

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

## Acknowledgements

We sincerely thank all members of the Miska lab for their valuable feedback on this work. We are grateful to the *Caenorhabditis* Genetics Center (CGC), funded by the National Institutes of Health Office of Research Infrastructure Programs (P40 OD010440), and the National Bioresource Project (NBRP) for providing the strains used in this research. We also extend our appreciation to Marie-Anne Félix's and Siu Sylvia Lee's labs for sharing worm strains. We are also thankful to Yuhua Lim and Lisa Lampersberger for their comments and support. This research was supported by grants from Wellcome Trust Senior Investigator Award (219475/Z/19/Z to E.A.M) and CRUK award (C13474/A27826 to E.A.M). Additional support was provided by grants from the National Natural Science Foundation of China (Grant 32200682 to J.X.). We also acknowledge core funding to the Gurdon Institute from Wellcome (092096/Z/10/Z, 203144/Z/16/Z) and CRUK (C6946/A24843).

## Author contributions

C.Z., G.F., M.V.A., J.C.R., H.S.R., J.X., E.C.S., and J.M. conducted the experiments. J.P., J.C.R., and M.V.A. analysed the sequencing data. M.V.A and P.R.-G. performed evolutionary and structural analysis of SET and SPK domains. J.X. analysed the images. K.Y. constructed the *C. elegans* strains. M.H. performed the western blotting under the supervision of S.G. F.B. processed samples and conducted mass spectrometry. E.A.M. supervised and funded the study. G.F. and E.A.M. conceived the study. C.Z. drafted the initial manuscript with contributions from coauthors, and all authors reviewed and edited the final version.

## Competing interests

The authors declare no competing interests.
