## [Transparent Peer Review file · Nature Communications]

A SET domain-containing protein and HCF-1 maintain transgenerational epigenetic memory

Corresponding Author: Professor Eric Miska

Version 0:

Reviewer comments:

Reviewer #1

(Remarks to the Author)

Overall thoughts

This paper identifies and characterizes a new gene (*set-24*) involved in transgenerational epigenetic inheritance. They start with a bioinformatic analysis of the gene, using a couple of phylogenetic approaches to suggest that its SET domain is most likely non-functional, and to identify two SPK domains. They next show that two different null mutants both display a mortal germline, and are defective in epigenetic inheritance – although the defect appears gradually over generations and is not as strong as their control, *hrde-1*. Using a GFP-fusion protein and qPCR expression analysis they show that SET-24 is present in the germline from the mitotic region to the pachytene. They perform some IP-MS experiments which suggest that HCF-1 interacts with SET-24, and go on to show that *hcf-1* mutants enhance TEI, rather than having a similar defect to *set-24*. Finally, they perform ChIP for H3K4me3, small RNA and mRNA seq on *set-24* mutants, and show a relatively small change in H3K4me3, which is associated with very minimal mRNA changes, and slightly more sRNA changes. They suggest that SET-24 modulates H3K4me3 and 22G levels at a small subset of genes.

The work is interesting and the claims made are substantiated by the experiments shown (for the most part, with some small exceptions outlined in the specific figure notes below), and will be of interest to the field. But I find that the research opens up a raft of further questions that must be answered to understand mechanistically how SET-24 functions.

How do you think HCF-1 interacts with SET-24? It doesn't have any HCF-1 binding motifs (Fig 1d), and you didn't identify SET-26 or SET-9 in your pull downs. It would be interesting to extend this analysis – for example: a pull down for SET-24 yielded HCF-1. If you pulled down HCF-1, do you get SET-24? What are the regions of SET-24 that are required to interact with HCF-1? Can you express SET-24 recombinantly and make various truncations to narrow down the region?

HCF-1 modulates HDACS, do you see changes in histone acetylation in SET-24 mutants?

Along a similar vein, you speculate that the SPK domains may bind dsDNA binding. It would be nice to test this hypothesis, again perhaps by using recombinantly expressed domains. What do you think that the SET domain is doing, if it isn't a methyltransferase – do you know if it's important to the function of SET-24, or are other parts of the protein functional instead. You could make mutants in worms and test them in your assays.

Given that HCF-1 and SET-24 display opposite phenotypes, yet you have evidence for interaction, what phenotypes do you see in a double mutant in your TEI and mrt assays?

The escapees in the mrt assay are fascinating and I was glad to see that you mentioned them in your discussion, and I enjoy your provocative alternative hypothesis. Have you considered characterising the escaped populations further? For example, if you take worms from the escaped population, or simply take the entire G10 cohort, and perform various omics analyses to compare them to the P0 population, what do you see? Changes in chromatin state, for example? Or small RNA populations? Performing these experiments would give a clearer idea of the mechanism by which SET-24 acts.

Specific notes on figures

Figure 1d It's a bit confusing to have *Chr11* written next to the gene structure, like the allele names are written next to the protein cartoons. Could that be moved elsewhere for clarity? Also, it's a minor point, but on my version the coloured boxes representing the protein domains in the gene locus are tiny and very hard to see. Perhaps this needs to be made a little bigger.

Figure S2C – this is interesting! Do you think that these can bind dsDNA? It would be good to express them recombinantly and test in some binding assays/gel shift experiments. Alternatively, could you use AlphaFold multimer to get some computational prediction of whether they may bind dsDNA?

Figure 2a/b It is hard to tell from the methods section and figure legend, whether these experiments were performed with

biological replicates (the experiment repeated independently on different days) or whether all the replicates were performed on the same day. As a small point, it might be worth pointing out in the legend or y axis of 1b that the offspring over only 3 days were counted, which is probably (hopefully) why there are less than expected in the wildtype control.

Fig 2c The images of the worms are lovely and clear and really make the point quite nicely... when they are enlarged. I wonder if you could shrink the quantification graphs and enlarge your representative images? Or otherwise rejig the figure, so that they can clearly be seen when printed.

Figure 3b In the set-24 strains, you have around ~90-100% GFP silenced worms in the G1 and in the G3 around ~50% of silenced worm. In the G3, you also had ~50% worms with a germline defect (Fig 2c). Can you confirm that the worms that had no GFP expression did actually have a germline, and that they aren't just GFP negative because they have no germline? I couldn't see this mentioned anywhere.

Figure 3c This is an interesting way to try and quantify the GFP expression. What were the RNA samples tested? Were the populations mentioned in the figure legend an entire plate at each generation, or a subset of a plate?

Fig 3d Nice! And interesting... At G1 neither set-24 allele is particularly defective in your visual GFP readout in 3b, nor different at the mRNA from WT in 3c, yet for set-24(mf123) at least, the 22G tested is almost at hrde-1 levels. At G4, set-24(mj617) appears quite defective at this 22G, but not particularly defective (yet) at the GFP level. There's a bit of a disconnect here, between 22Gs supposed roles in heritable RNAi and these results, suggesting it's not quite so clear cut. I don't dispute the summary of the findings in lines 173-175, but perhaps the implications of them aren't so clear cut.

Figure 4 Lovely imaging. Given the creation of the GFP tagged strains, are the qRTPCRs in a&b really necessary to include in the main figure? Perhaps instead, you could look more closely at the overlap between SET-24::GFP and chromatin? It looks like the DAPI signal and GFP signal overlaps well, but not completely. Could you do some analysis to determine the amount of overlap/chromatin occupation? What about overlap with the X chromosome, is it consistently excluded (for example)? You also show quite convincingly in the qPCR experiments that the expression of SET-24 is mostly germline specific, but nonetheless, you do see some expression in the glp-4 mutants. It could be useful to look at imaging of the whole worm to look at potential other expression of SET-24 (i.e. is it expressed in neurons?)

Related to Figure 5 – around Line 250-252 you should probably also include a reference to the Lee lab's recent paper on SET-26 and HCF-1 (already cited elsewhere as ref 61) because this paper has pretty good evidence suggesting a direct interaction between SET-26 and HCF-1

Figure 6 Here you've focused on protein coding genes, but do you see any changes globally? For example, if you call peaks in your WT and set-24 experiments, do they mostly overlap? Given the subtle increase in H3K4me3 signal in set-24 mutants, perhaps you see an increase in peaks at non-protein coding regions of the genome too?

It is difficult to understand how the analysis for determining H3K4me3 enriched genes has been carried out. From reading the methods (lines 685-686) and figure legend, it seems that the H3K4me3 enriched genes have been determined based on enrichment over input in WT (i.e. the genes that are targeted by this mark in WT), and then this subset is shown to be further enriched in H3K4me3 in set-24 mutants. However, the text in lines 267-269 suggests that the H3K4me3 enriched genes (note typo in fig 6a top panel) enrichment was determined in set-24 mutants. Can this please be clarified?

The ChIP/mRNAseq/sRNA is performed in whole worms, while SET-24 is expressed in the germline, thus presumably diluting/obscuring the signal. Could you perform these experiments on dissected germlines instead? Or could you take an alternative approach and look for changes in H3K4me3 in the germline by IF?

Typo in 6f – I think the middle panel should read CPM (mRNA) and the bottom panel should be CPM (22G-RNA)

Reviewer #2

(Remarks to the Author)

In this manuscript the authors describe a function for SET-24 in the maintenance of transgenerational epigenetic inheritance (TEI) through the balance of DNA methylation and small RNA production. They emphasize two major findings that distinguish SET-24 from previously described TEI factors: 1) SET-24 is a methyltransferase that lacks both the canonical catalytic motif and the canonical DNA-binding PHD motif and 2) SET-24 is specifically required for maintenance but not for establishment of TEI responses.

Major revisions are needed, specifically to strengthen the link between SET-24 and its actions on chromatin/DNA and to clarify how SET-24's impacts on chromatin/small RNA contribute to the Mrt TEI phenotype.

Major points:

Unfortunately, how SET-24 interacts with DNA seems murky at this point, a major missing link. Not only would an understanding of this interaction bridge the TEI phenotype to the downstream H3K4me3 and small RNAs mechanisms, it also would provide a new perspective concerning how a catalytically inactive methyltransferase can act to influence chromatin state. The authors themselves propose a likely scenario –the SPK (MYB-like) domains participate in ds-telomeric DNA binding. The authors should (1) delete/mutate the SPK domains site-specifically and assay TEI; (2) delete/mutate the SPK domains site-specifically and check for SET-24 mislocalization from DNA regions using their fluorescently-tagged SET-24 strain; and/or (3) add to their AlphaFold data and look at potential DNA-binding conformations of SET-24.

Related to the above issue, SET-24 might be associating with HCF-1 to bind DNA (even though they are functional opposites phenotypically for TEI). Clarity about how SET-24 and HCF-1 affect each other besides being binding partners should be addressed. Human MLL5 and HCFC1 have previously been shown to be interacting partners, and this fact should be stated explicitly. The relevant citation, #78 is in the References but seems to be describing something else in the text. set-24(lf) and hcf-1(lf) do not affect each other's localization as visualized. We recognize that the current reagents for localizing both SET-24 and HCF-1 are both GFP-based, but the 5th paragraph of the main text is subtitled "The SET-24 interactor, HCF-1 ..." and it is important to bolster this claim by one or more of the following experiments: (1) Imaging both SET-24 and HCF-1 at higher resolution to check for co-localization, which should be easily done, perhaps by imaging the HCF-1::GFP

strain with an antibody label (using a separate non-GFP green color) against the FLAG on the SET-24::3xFLAG and with DAPI to visualize the DNA. (2) SET-24 and HCF-1 are proposed to be interacting partners, yet they are shown to act in opposing directions for the TEI phenotype. To clarify how this interaction is regulated, a double mutant should be constructed and tested for TEI or signs of epistasis in the pathway. (3) The authors might also check for the previously identified (in human) HCF-1-binding motif (HBM) site in SET-24 as mentioned in reference #78 and perhaps give more insight into how that motif fits in their AlphaFold models and implications for how the potential HCF-1 binding site interacts structurally with the SET-24 SPK domains that supposedly bind DNA (e.g., do they occlude each other sterically, does the structural orientation of the domains inform how the SET-24-HCF-1-DNA complex might work etc.). (4) An in vitro interaction assay, e.g., if purified SET-24 (using the FLAG IP) were found to bind HCF-1::GFP (GFP Ab IP) together with purified DNA that would provide evidence for a SET-24-HCF-1-DNA interaction.

The H3K4me3 enrichment and any resultant effects on transcription in *set-24(lf)* mutants compared to wild-type worms appear mild. To support their claims that altered H3K4me3 levels and thus chromatin state is what SET-24 is affecting to regulate TEI, the authors should compare the levels of H3K4me3 and 22-G RNAs in the G3 *set-24(lf)* affected lines with those in the "escapee" wild-type-looking lines at G4 to see if H3K4me3 and 22-G RNA levels return to normal. This experiment might link proposed downstream molecular mechanistic effectors of 22G-RNA levels to the organismal Mrt phenotype with good (generational) temporal specificity. Such a finding would be of considerable interest given the proposed adaptive biological functions of SET-24 discussed in the last paragraph of the paper. Relatedly, such an experiment could also be done with HCF-1 mutants, which show opposing TEI effects, to see if the H3K4me3 and 22G-RNA changes identified in *set-24(lf)* mutants are corresponding opposed in *hcf-1(lf)* mutants thus demonstrating mechanistic bidirectionality in tying the phenotypic observations of TEI to the molecular players.

Minor points:

The authors should explain why the wild-type responses for depressed GFP-RNAi differ quite dramatically between Fig 3b (<10-20% at G7) and Fig 5d (close to 100% at G7)?

The authors have presented a comprehensive characterization of the phenotypic TEI consequences of the loss of SET-24. The authors make a case of the striking G3 vs G4 Mrt difference, attributing it to "escapee" takeover of the population. There should be a more detailed population breakdown or description of this rather sharp transition reversal. If one were to pick Mrt-displaying adult worms with degenerated germlines in G3 and assay their G4 progeny what would be the outcome? Would all G4 progeny be sterile/non-viable? As it stands the increase in animals having normal germlines past G4 might be due to reversion or compensation mechanism being activated rather than "escapee" takeover and these scenarios could be distinguished.

SET-24's germline-specific expression was previously established (albeit only for L1 stages) by Engert et al. (2018) (doi:10.1371/journal.pgen.1007295). Given the emphasis on SET-24 expression in this submitted manuscript, this fact should be explicitly stated and this earlier paper should be cited.

Are any of the identified WAGO and Mutator genes identified in this paper possible effectors of the Mrt phenotype? Have they appeared in screens / sequencing from studies involving the Mrt phenotype?

Reviewer #3

(Remarks to the Author)

In this manuscript from Zeng et al, the authors examined a SET-domain containing protein (SET-24) that plays roles in germline immortality and TEI in *C. elegans* according to their previous published work. The SET domain is typically the catalytic domain of histone methyltransferase. However, the authors showed that SET-24 shares sequence similarity with the scSET3 superfamily members that are catalytic inactive. Therefore, whether SET-24 can methylate histone is unclear. Interestingly, SET-24 is expressed in germline lineage of the *C. elegans*, and is required for germline immortality and heritable RNAi, suggesting it plays important roles in epigenetic inheritance. Using IP-Mass Spec and yeast two-hybrid assay, the authors show that SET-24 interact with HCF-1, a key component of the COMPASS chromatin remodeling complex that regulate H3K4me3. The author then examine the role of SET-24 in regulating H3K4me3, mRNA and small RNA expression. In *set-24* mutant, a mild increase of H3K4me3 is observed at a subset of transcription start sites(TSS), but few corresponding mRNA changes were observed. In contrast, more small RNA changes are seen at these TSS, and these small RNAs belong to the WAGO and Mutator regulated genes.

Overall, the manuscript provides strong evidence that SET-24 is required for germline immortality and TEI, offering valuable insight into the field of epigenetic inheritance. However, the modest H3K4me3 changes at a subset of TSS in the *set-24* mutant also raise the questions about whether these loci are direct targets of SET-24/HCF-1 (COMPASS complex). The genome-wide analysis (ChIP-seq, mRNA-seq, small RNA-esq) lacks information regarding biological replicates and the allele used and requires major revision.

Specific comments:

1. The structure-based phylogenetic analysis supports that key residues involving in catalytic activity in the SET domain are missing in SET-24, SET-9, and SET-26. However, SET-26 has also been reported to have H3K9 HMT activity in vitro (PMID: 24685137). It remains possible that these SET domains utilize alternative residues for catalytic function.
2. The SET-domain of SET-24 appears to end before the 144 a.a. (Fig. 1C, Fig. S1). The *set-24 mf123/mj617* allele is truncated after the 188 a.a. (Fig. 1D). It is not clear if the SET-domain remained in this mutant.
3. Fig. 4C: SET-24 form puncta in the embryo P-lineage. Some SET-24 puncta appear to be cytoplasmic and is enriched in PGL-3 foci.
4. Line 217-222, Fig. S6: ChIP might still work even without clear DNA enrichment. This could be due to higher background (since SET-24 only expressed in a subset of germline line cells) or fewer binding sites. Sequencing these samples might still reveal the binding sites.
5. The detection of HCF-1 in both the IP-MS and Y2H supports an interaction between SET-24 and HCF-1. How many biological replicates were performed for these experiments? Were other COMPASS complex components identified in the IP-

MS? Including a co-IP validation of SET-24 and HCF-1 would further strengthen the conclusion.

6. Line 238, Fig5: Is the hcf-1(ok559) mutant a null mutant? More information on this allele is necessary to interpret HCF-1's role in the fertility assay and heritable RNAi assay.

7. Which of the two set-24 mutants were used for the H3K4me3 ChIP-seq? Do both set-24 alleles exhibit the same changes in H3K4me3 levels? How many biological replicates were included and how are they? The observed H3K4me3 increase at a subset of TSS of set-24 mutant is modest (<20%, based on Fig 6b), raise the question of whether these TSS sites are the direct or indirect target of the SET-24/HCF-1 (COMPASS complex).

8. Similarly, the mRNA-seq and small RNA-seq analyses lack details about which set-24 allele was used and how many biological replicates were performed.

9. Since SET-24 is germline-specific, how many of the 757 genes showing SET-24-dependent H3K4me3 are also germline expressed? This analysis could help support a direct role for SET-24 in regulating H3K4me3 via HCF-1. Additionally, do the genes with altered small RNA levels in the set-24 mutant also fall into the category of germline-expressed genes?

Minor comments:

Line 178: "transcript sand" should be "transcripts and".

Figure 6f: Can not find track labels for sRNA-seq or mRNA-seq.

Reviewer #4

(Remarks to the Author)

Version 1:

Reviewer comments:

Reviewer #1

(Remarks to the Author)

I would like to commend the authors on the large amount of extra work that they have performed and included in this revised manuscript. Below are some comments on the additions, in the context of what I wrote in my previous review.

Given the increased focus on the opposite effect of HCF-1 and SET-24, is the title still appropriate? It does make it sound as though they act in the same way, when actually they have opposite effects on epigenetic memory.

I'm also still not convinced that the authors have really nailed the mechanism by which SET-24 acts to regulate TEI.

My previous main points

1. How does SET-24 interact with HCF-1? The authors have added a lot of new data to the manuscript to address this point, and have shown that the Kelch domain of HCF-1 interacts with the SET domain (and its immediate flanking regions) of SET-24. They furthermore show that loss of one protein does not affect the localization of the other, using fluorescent reporter strains.

2. Do they see changes in histone acetylation in set-24 mutants? They did not see any global changes to histone acetylation (although it's worth noting that they also didn't see global changes to H3K4me3 by western blot).

3. Do the SPK domains of H3K4me3 bind DNA as they speculate? The authors recombinantly express the SPK domain and tested them in a band shift assay, but did not see any evidence of DNA binding. I appreciate the effort, and acknowledge that absence of evidence is not evidence of absence, so leaving the comment in the Discussion (lines 445-446) is probably OK. I still do think that answering this question would be informative for the mechanism.

4. What is the SET domain doing? They showed that the SET domain seems to be required for stability of SET-24, as mutations in the SET domain in worms could no longer be visualized with the GFP reporter, and protein could no longer be detected by western blot (although mRNA could). This is a good advance to the paper.

5. What are the phenotypes of hcf-1::set-24 double mutants? They show that in all cases tested (not just the epigenetic inheritance assay), double mutants phenocopy hcf-1 mutants, and propose that this is due to HCF-1 having broader roles with other chromatin remodelling complexes. I think these results really highlight the fact that although SET-24 and HCF-1 seem to be interacting, it's really not clear how this fits into the broader context of TEI.

6. What is going on with the escapees? I suggested various 'omics analyses could be performed. The authors did some small RNA seq on one population of escapees and showed dysregulation of small RNAs – not included in the manuscript. They also suggest that other experiments are ongoing and will form the basis of a later manuscript. I'm left a bit unsatisfied, but I guess this is OK on balance, give the amount of other data that they've added.

7. Does SET-24 colocalise with chromatin? The authors have addressed this well by doing colocalization analysis with H2B::mCherry and tagRFP::SYP-1, and show that SET-24 only partially localises with chromatin. Worth pointing out is the apparent expression of SET-24 in granules (overlapping partially with PGL-3) in the P3 or P4 cells. This is non-nuclear (although still germline)! The authors should comment on this in the text. It might be meaningful for the action of SET-24 in the context of small RNA regulation.

Other comments

1. The recombinant proteins used for the in vitro methyltransferase assays are not at all well purified. There are many contaminants in the Coomassie gel (S1a). Can the authors provide more information to prove that that have indeed successfully expressed MBP-CeSET-24 and that it is folded correctly? If it isn't folded correctly (or expressed well) this would explain the negative result.

Reviewer #2

(Remarks to the Author)

I am satisfied with the alterations the authors have made to the paper and accept their explanations for the limitations in certain proposed experiments as either technically or otherwise beyond the scope of the paper.

I appreciate the increased clarity about the characterization of SET-24, its strength of interaction with HCF-1 and analysis of the HCF-1 TEI phenotype.

I have the following specific additional minor comments:

1) It seems the proposed model is that SET-24 and HCF-1 are acting antagonistically (and coordinately) in a balance to maintain proper (wild-type) heritable RNAi generational duration. In lines 422-423, for example, it would be both clearer and more accurate to explicitly spell out the molecular-phenotypic description with appropriate directionality rather than just saying "required for heritable RNAi," which could easily be read as HCF-1's acting in the same direction as SET-24.

2) SET-24 and HCF-1 act antagonistically in contributing to the TEI phenotype, and *hcf-1* single mutants and *hcf-1; set-24* double mutants are equivalent in phenotype, i.e. *hcf-1* shows "epistasis" not "dominance" to *set-24* (line 335). The authors might make explicit in that context the statement about *hcf-1*'s being downstream of (or parallel to) *set-24* in mediating the TEI phenotype (similar to the statement in lines 376-377, "suggesting a role for SET-24 in ... regulation through HCF-1 ..."), a conclusion that would be a standard interpretation of epistasis. They might further juxtapose those statements with one about the observation that the two are in the same physical complex, and conclude that part of the discussion with their proposed molecular model of how the two could act antagonistically while being in a physical complex. Such clarity would help contextualize the authors' findings and make a general statement that would highlight the discovery made.

3) In lines 535 onwards, could the authors clarify their specific hypothesis about the proposed advantage of a mutant *set-24* allele in a wild isolate? A straightforward explanation would be that the Mrt phenotype is an epi-phenomenon and that there is a selective advantage under certain environments to some other consequence of reduced *set-24* function. The second hypothesis (line 547) seems to be a more complicated and perhaps unnecessary take on this general explanation. Also, the first hypothesis/explanation (line 537) could explain the Mrt "escapees"/reversibility, but I did not understand why Mrt would be beneficial in the first place? If anything, Mrt has a strong direct negative impact on fertility.

4) In line 306, the authors should clarify "out-of-frame" mutation. "out-of-frame" is not a standard genetic phrase and might be interpreted as "frameshift," "deletion or insertion causing a frameshift" or other. *ok559* seems to be a deletion mutation that results in an early stop codon (and frameshift?). Whatever the nature of *ok559*, it should be clearly specified in those terms.

Reviewer #3

(Remarks to the Author)

In this revision, the authors experimentally demonstrated that SET-24 lacks detectable HMT activity, and provided additional evidence for the interaction between SET-24 and HCF-1, as well as the domains required for this interaction. The sequencing analysis has been further strengthened and clarified. Overall, the manuscript's conclusions are substantially stronger, and my previous concerns have been satisfactorily addressed.

Reviewer #4

(Remarks to the Author)

Response to reviewers:

Reviewer #1 (Remarks to the Author):

Overall thoughts

This paper identifies and characterizes a new gene (set-24) involved in transgenerational epigenetic inheritance. They start with a bioinformatic analysis of the gene, using a couple of phylogenetic approaches to suggest that its SET domain is most likely non-functional, and to identify two SPK domains. They next show that two different null mutants both display a mortal germline, and are defective in epigenetic inheritance – although the defect appears gradually over generations and is not as strong as their control, hrde-1. Using a GFP-fusion protein and qPCR expression analysis they show that SET-24 is present in the germline from the mitotic region to the pachytene. They perform some IP-MS experiments which suggest that HCF-1 interacts with SET-24, and go on to show that hcf-1 mutants enhance TEI, rather than having a similar defect to set-24. Finally, they perform ChIP for H3K4me3, small RNA and mRNA seq on set-24 mutants, and show a relatively small change in H3K4me3, which is associated with very minimal mRNA changes, and slightly more sRNA changes. They suggest that SET-24 modulates H3K4me3 and 22G levels at a small subset of genes.

The work is interesting and the claims made are substantiated by the experiments shown (for the most part, with some small exceptions outlined in the specific figure notes below), and will be of interest to the field. But I find that the research opens up a raft of further questions that must be answered to understand mechanistically how SET-24 functions.

Thank you for your comments. Your suggestions have greatly helped us improve our manuscript.

How do you think HCF-1 interacts with SET-24? It doesn't have any HCF-1 binding motifs (Fig 1d), and you didn't identify SET-26 or SET-9 in your pull downs. It would be interesting to extend this analysis – for example: a pull down for SET-24 yielded HCF-1. If you pulled down HCF-1, do you get SET-24? What are the regions of SET-24 that are required to interact with HCF-1? Can you express SET-24 recombinantly and make various truncations to narrow down the region?

In the revised manuscript, we address this point with additional experimental evidence (in Figs. 5, S6, and S7) further dissecting the SET-24-HCF-1 interaction and textual changes (please see lines 268-270, 285-301, and 464-471).

In the previous manuscript, we showed by IP-MS and yeast two-hybrid (Y2H) assays that HCF-1 and SET-24 interact. Now, we have conducted additional *in vitro* pull-down assays (please see **Figs. 5c** and **S6c**) that **confirm a direct physical interaction between full-length SET-24 and HCF-1**.

Figs. 5c and S6c

We went further and mapped the interactions to specific protein domain, using *in vitro* pull-down assays. These approaches revealed that residues 1–378 of HCF-1 (mainly containing the Kelch domain) directly interact with SET-24 (**Figs. 5f, 5h, and S7a**). All SET-24 truncations containing the N-terminal 1–239 region, which includes the SET domain and its flanking sequences, bound to the HCF-1 1-378 fragment (**Figs. 5g, 5i, S7b, and S7c**). Interestingly, the SET domain and some of its flanking sequences seem to be necessary for association with HCF-1 (**Figs. 5g, 5j, 5k, S7d, S7f**).

Together, these data suggest **the Kelch domain of HCF-1 directly interacts with the SET domain of SET-24, with the flanking sequences of the SET domain also contributing to the interaction.**

HCF-1 modulates HDACS, do you see changes in histone acetylation in SET-24 mutants?

Although HDA-1 was pulled down by HCF-1 in IP-MS reported by Emerson et al.¹, our IP-MS and Y2H assays did not identify any HDACs as interactors of SET-24. We assessed global histone acetylation levels through western blotting on wild-type, *set-24* mutants, and *hcf-1* mutants. No significant changes in global histone acetylation levels were observed in the mutants (**Fig. S9b**). Thus, the SET-24-HCF-1 regulatory nexus may work independently of HCF-1's interactions with HDA-1. We propose that testing whether HDACs participate in the SET-24-HCF-1 regulatory nexus is beyond the scope of this manuscript.

Along a similar vein, you speculate that the SPK domains may bind dsDNA binding. It would be nice to test this hypothesis, again perhaps by using recombinantly expressed domains.

Thank you for your comments. To better address the comment, we have generated a figure for the reviewer only (see the **Revision Fig. 1** below). To address this, we established a band shift assay using a DNA-binding protein (the tandem helix-turn-helix domains of the Mos1 transposase) and its known DNA target, as positive control^{2,3}. In this case, a clear DNA band shift is observed (**Revision Figs. 1a and 1b**). We then expressed and purified SPK domains *in vitro* (**Revision Figs. 1c and 1d**). As the potential direct DNA targets of the SPK domains remain unknown, we tested the ability of the SPK domains to bind to several loci exhibiting H3K4me3 changes in *set-24* mutants (as defined in Fig. 7a and Supplementary table 4; genomic regions amplified by PCR), and additionally to a DNA library containing random sequences of the same size. We did not observe band shift (**Revision Figs. 1e-1g**). This suggests that the SPK domains in SET-24 may not bind DNA directly. Alternatively, the DNA sequences tested may not be direct targets.

Developing an experimental system to systematically investigate DNA binding by SPK, an uncharacterized protein fold with a yet unknown potential DNA target, falls beyond the timeline of this revision and the scope of the paper.

Revision Fig. 1 (for the reviewer only) **a** Coomassie blue staining of purified GST and GST-MOS1-HTH, the two HTH domains of the transposase Mos1 that have known DNA-binding ability. **b** Electrophoretic mobility shift assay (EMSA) demonstrating high-affinity binding of the isolated Mos1 transposase double-HTH domain to its cognate terminal inverted repeat DNA sequence. **c** Schematic diagram of the GST-tagged constructs with SPK domains of SET-24. **d** Coomassie blue staining of the purified GST-tagged SPK domains. **e** EMSA of the SPK domains with a mixture of 200 bp DNA fragments from H3K4me3-enriched genes. **f** Schematic of a double-stranded nucleic acid pool containing random sequences of fixed length. The single-stranded DNA library, containing primer binding regions at the 5' and 3' ends, was synthesized by IDT. The double-stranded DNA pool was then generated by PCR amplification using primers complementary to the binding regions and using the single-stranded DNA library as template. **g** EMSA of the SPK domains with the random double-stranded DNA pool. The DNA-binding protein MOS1-HTH did

not show a band shift with the double-stranded DNA pool, likely due to the insufficient representation of the target sequence in the random-sequence DNA pool.

What do you think that the SET domain is doing, if it isn't a methyltransferase – do you know if it's important to the function of SET-24, or are other parts of the protein functional instead. You could make mutants in worms and test them in your assays.

This is an important question. To address it, the manuscript now includes additional experimental data and textual changes (**Figs. 4e-4g, 5f-5k, S5h-S5i, and S7**, please see lines **233-253 and 285-301**).

As described above, in the revised version of the manuscript we mapped the regions of SET-24 and HCF-1 mediating their interactions (**Figs. 5f-5k and S7**). The SET domain of SET-24 interacts with Kelch domain of HCF-1 *in vitro*, which suggests a structural role for the SET domain, by providing a platform for interactions with other proteins.

Then, as you suggested, We generated SET domain mutants in both *set-24::gfp* and *set-24::3xflag* strains (**Figs. 4e and S5h**). Specifically, *mj673* and *mj677* introduce a five–amino acid substitution within the conserved region of the ScSET3 (mutating RRSCQNN to ASASQNA) in *set-24::gfp* and *set-24::3xflag* strains. The *mj675* mutation deletes a sequence spanning both intronic and exonic regions, but remains in-frame, whereas *mj674* and *mj676* introduce frame-shift deletions in *set-24::gfp* and *set-24::3xflag* strains, respectively (**Figs. 4e and S5h**).

qPCR analysis revealed that *set-24* mRNA levels in the point mutants (*mj673* and *mj677*) were comparable to wild-type, whereas the *mj675* deletion allele showed reduced abundance, likely due to loss of regulatory sequences in an intron (Fig. S5i). Crucially, although *set-24* mRNA levels remain unchanged in *set-24(mj673)::gfp* and *set-24(mj677)::3xflag* strains, SET-24 protein was undetectable in all three mutants (**Figs. 4f and 4g**). This finding indicates that the SET domain, including a set of residues highly conserved in the ScSET3 family, is required for stability of SET-24 protein. The absence of protein in these mutants implies that they would behave similar to *set-24(mj617)* and *set-24(syb7014)* in Mrt and TEI assays (*mj617* is a null allele, and *syb7014* phenocopies *mj617*).

We conclude that the SET domain of SET-24 has a structural role, stabilising SET-24 protein and functioning as a platform for interactions with other proteins.

Figs. 4f and 4g

Given that *HCF-1* and *SET-24* display opposite phenotypes, yet you have evidence for interaction, what phenotypes do you see in a double mutant in your TEI and *mrt* assays?

In the revised manuscript, we provide additional experimental data addressing this (please see **Figs 6a, 6e-6g, and S8c-S8f, and lines 319-333**).

Both *hcf-1* mutants reduced brood size of wild-type worms at 20 °C and are nearly sterile at 25 °C within one or two generations, without producing “escapees” (**Fig. 6a**). This phenotype contrasts with that of *set-24* single mutants at both 20 °C and 25 °C (**Fig. 6a**). To answer the reviewer’s comment, we examined the brood size of *set-24; hcf-1* double mutants and found that their progeny numbers were comparable to those of the *hcf-1* single mutants (**Fig. 6a**).

Fig. 6a

In addition, we quantified the inheritance of RNAi silencing of a GFP reporter in *set-24; hcf-1* double mutants. We silenced heterozygous *+/set-24; +/-hcf-1* worms carrying the GFP::H2B transgene by feeding *gfp* dsRNA and monitored silencing inheritance in their progeny (including a range of genotypes: wild-type, *set-24* or *hcf-*

1 single mutants, and *set-24*; *hcf-1* double mutants; see Fig. 6e). Consistent with prior observations, silencing was lost more quickly in *set-24* single mutants than in wild-type animals, whereas it was maintained for more generations in *hcf-1* single mutants (Figs. 6f and 6g). Importantly, *set-24*; *hcf-1* double mutants also maintained silencing longer than wild-type worms, similar to the *hcf-1* single mutants (Figs. 6f and 6g). This phenocopying was observed when experiments were repeated using a different genetic experimental design (Figs. S8c-S8f).

Figs. 6e-6g

We conclude that *set-24*; *hcf-1* double mutants phenocopy *hcf-1* single mutants in terms of fertility defects and RNAi inheritance. We propose that the opposite phenotypes of *set-24* and *hcf-1* mutants, and the dominance of the *hcf-1* mutant phenotype are due to the broader regulatory roles of HCF-1 in concert with multiple chromatin remodelling complexes.

The escapees in the mrt assay are fascinating and I was glad to see that you mentioned them in your discussion, and I enjoy your provocative alternative hypothesis. Have you considered characterising the escaped populations further? For example, if you take worms from the escaped population, or simply take the entire G10 cohort, and perform various omics analyses to compare them to the P0 population, what do you see? Changes in chromatin state, for example? Or small RNA populations? Performing these experiments would give a clearer idea of the mechanism by which SET-24 acts.

Thank you for your comment. We fully agree that the “escapee” phenotype in the Mrt assay is very intriguing and worth studying. We believe the occurrence of “escapees” is an epigenetic effect. Investigating this phenomenon will not only help reveal the mechanism by which SET-24 functions, but also provide broader insights into how worms can escape environmental stress (high temperature) through epigenetic regulation.

As you suggested, we have profiled small RNAs in P0 and escaped populations. We sequenced 22G-RNAs in wild-type and *set-24(syb7014)* mutant worms at 25 °C at both P0 and G12 generations (the latter expanded from a single G10 “escapee”; four replicates each condition). The 22G-RNA population at G12 (*set-24* vs. wild type) differs from that at P0 under 25 °C conditions (**Revision Figs. 2a-2c**), suggesting that 22G-RNA deregulation plays a role. The genes with deregulated 22G-RNA levels overlap partially between P0 and G12, in some cases even changing regulation (i.e. from down- to upregulated, or vice-versa; see Revision **Fig. 2c**). Thus, the small RNA deregulation associated with escape of Mrt phenotype is complex.

Revision Fig. 2 (for the reviewer only) a Plots showing the proportion of small RNAs length distribution profiles, along with information on the first nucleotides in wild-type and *set-*

24(*syb7014*) samples at 25°C P0 or G12 (“escapees” for *set-24* mutants), which were treated with RppH. **b** Venn diagrams showing overlap of targets of differentially expressed 22G-RNAs in *set-24*(*syb7014*) at 25°C P0 and G12 (fold change > 2 and 5% FDR), G12 worms were collected from the “escapees” lines. **c** Upset plot showing overlap of targets of upregulated- and downregulated 22G-RNAs in *set-24*(*syb7014*) at 25°C P0 and G12.

Because of the apparent complexity of the deregulation and stochasticity of the phenotype, additional comprehensive analyses are warranted, for example sequencing small RNAs on a substantially larger number of biological replicates of escapee populations. However, we believe these could be more effectively tackled in a dedicated study. Indeed, we are planning future studies to investigate how “escapees” arise in *set-24* mutants growing in high temperature conditions. Given the experimental caveats stated above, we plan to perturb the expression of a series of epigenetic factors involved in chromatin remodelling and small RNA pathways during a Mrt experiment. This approach may help identify which pathways, when disrupted, can block the occurrence of “escapees”.

Specific notes on figures

Figure 1d It’s a bit confusing to have *ChrII* written next to the gene structure, like the allele names are written next to the protein cartoons. Could that be moved elsewhere for clarity? Also, it’s a minor point, but on my version the coloured boxes representing the protein domains in the gene locus are tiny and very hard to see. Perhaps this needs to be made a little bigger.

Thank you for your comments. We have removed the “*ChrII*” label and enlarged the coloured boxes in **Fig. S3a**.

Figure S2C – this is interesting! Do you think that these can bind dsDNA? It would be good to express them recombinantly and test in some binding assays/gel shift experiments. Alternatively, could you use alphafold multimer to get some computational prediction of whether they may bind dsDNA?

As mentioned earlier, we tested this hypothesis using the available materials and techniques but did not detect direct DNA binding (see **Revision Fig. 1**). We also used AlphaFold3 to model potential interactions between the SPK domains of SET-24 and telomeric repeats, which are typically recognized by related telomere-binding MYB domains. However, no reliable interactions were observed, possibly due to the DNA-binding sequence of MYB domains is not an appropriate proxy for the SPK domains.

Figure 2a/b It is hard to tell from the methods section and figure legend, whether these experiments were performed with biological replicates (the experiment repeated independently on different days) or whether all the replicates were performed on the

same day. As a small point, it might be worth pointing out in the legend or y axis of 1b that the offspring over only 3 days were counted, which is probably (hopefully) why there are less than expected in the wildtype control.

Figure 2a/b were generated from independent experiments performed on different days. We are now clearly specifying this in the figure legend (please see the **figure legend of Fig. 2**)

The y-axis shows the total number of offspring **at 25 °C**, and the brood size is consistent with other results (e.g., brood size in Fig. 6a). To make this clearer, we have labelled the axis with “25 °C.” Offspring counting began once the progeny reached the L4 stage, and counted offspring were removed using a vacuum pump. Over three consecutive days, all offspring from a single worm were counted. We included this information in the methods section (please see lines **634-635**).

Fig 2c The images of the worms are lovely and clear and really make the point quite nicely... when they are enlarged. I wonder if you could shrink the quantification graphs and enlarge your representative images? Or otherwise rejig the figure, so that they can clearly be seen when printed.

That is a good point. We have enlarged the representative images in **Fig.2c**.

Figure 3b In the set-24 strains, you have around ~90-100% GFP silenced worms in the G1 and in the G3 around ~50% of silenced worm. In the G3, you also had ~50% worms with a germline defect (Fig 2c). Can you confirm that the worms that had no GFP expression did actually have a germline, and that they aren't just GFP negative because they have no germline? I couldn't see this mentioned anywhere.

The silencing assay was performed at 20 °C, whereas the Mrt assay was conducted at 25 °C. At 20 °C, the brood sizes of *set-24* mutants were comparable to those of wild type (**Fig. 6a**), and the germline appeared generally normal. Consequently, the silenced worms in generation 3 (G3, 20 °C) retained a germline.

Figure 3c This is an interesting way to try and quantify the GFP expression. What were the RNA samples tested? Were the populations mentioned in the figure legend an entire plate at each generation, or a subset of a plate?

For wild-type and *set-24* worms, we picked one repressed worm at each generation (when available) and extracted RNA from its progeny (entire plate), after setting aside one GFP-silenced worm to propagate the next generation. As indicated in the figure legend, five populations (from five plates) were used at each generation as biological replicates

for each genotype. For the *hrde-1* mutant, we picked derepressed worms starting from G1. This information can be found in the methods section (see lines 657-660).

Fig 3d Nice! And interesting... At G1 neither set-24 allele is particularly defective in you visual GFP readout in 3b, nor different at the mRNA from WT in 3c, yet for set-24(mf123) at least, the 22G tested is almost at hrde-1 levels. At G4, set-24(mj617) appears quite defective at this 22G, but not particularly defective (yet) at the GFP level. There's a bit of a disconnect here, between 22Gs supposed roles in heritable RNAi and these results, suggesting it's not quite so clear cut. I don't dispute the summary of the findings in lines 173-175, but perhaps the implications of them aren't so clear cut.

That's a very good point. We propose this is likely related to our observations showing that SET-24 affects both chromatin modifications and small RNA biogenesis (**Figs 7 and S11**). Our data suggests that SET-24 contributes to the maintenance of RNAi-induced silencing through regulation of 22G-RNAs, but the slight disconnection between 22G-RNA and GFP expression is likely due to the balance of 22G-RNAs plus the chromatin landscape at the GFP reporter locus. We note that the variability between the *set-24* alleles may also reflect slight stochastic differences in levels of 22G-RNAs targeting *gfp* and chromatin landscape deregulation at the GFP reporter locus. We think this stochastic variability may have implications in the emergence of escapees at higher temperatures. As discussed above, a deep comprehensive drive into the basis of escapee emergence falls beyond the scope of this paper.

*Figure 4 Lovely imaging. Given the creation of the GFP tagged strains, are the qRTPCRs in a&b really necessary to include in the main figure? Perhaps instead, you could look more closely at the overlap between SET-24::GFP and chromatin? It looks like the DAPI signal and GFP signal overlaps well, but not completely. Could you do some analysis to determine the amount of overlap/chromatin occupation? What about overlap with the X chromosome, is it consistently excluded (for example)? You also show quite convincingly in the qPCR experiments that the expression of SET-24 is mostly germline specific, but nonetheless, you do see some expression in the *glp-4* mutants. It could be useful to look at imaging of the whole worm to look at potential other expression of SET-24 (i.e. is it expressed in neurons?)*

Thank you for your comments. We have moved the qRT-PCR data to the supplementary figures (**Fig. S5a and S5b**). We agree that examining the overlap between SET-24::GFP and H2B is an excellent idea. Using confocal microscopy, we analysed the colocalization of SET-24::GFP and H2B::mCherry in the live worms by calculating Pearson's colocalization coefficient in ImageJ. This analysis showed that SET-24::GFP partially colocalizes with H2B::mCherry in the nucleus (**Fig. 4b and 4d**). Conversely, the colocalization between SET-24::GFP and tagRFP::SYP-1, a marker of the synaptonemal

complex, is weaker (**Fig. 4c and 4d**). With this resolution and by imaging live worms, we detected SET-24::GFP signal dispersed throughout germline nuclei, but also within puncta. The SET-24 puncta did not colocalize well with H2B::mCherry and tagRFP::SYP-1, suggesting that they are not associated with chromosomes (**Figs. 4b and 4c**). This suggests that there are two pools of SET-24 protein in germline nuclei: one more closely associated with chromatin, and another forming distinct puncta, which do not overlap chromosomes. We included this data in the revised manuscript (please see **Figs. 4b-4d**) and adjusted text accordingly (please see lines **223-231**).

Following the reviewer's comment, we also revisited SET-24 expression beyond the germline. Although a small fraction of SET-24 mRNA is detected in germlineless mutants, examination of SET-24::GFP expression throughout the whole worm with an increased resolution confirmed that SET-24 is specifically expressed in the germline (Please see **Fig. S5f** and line **216**). Specifically, SET-24::GFP is not expressed in neurons.

Related to Figure 5 – around Line 250-252 you should probably also include a reference to the Lee lab's recent paper on SET-26 and HCF-1 (already cited elsewhere as ref 61) because this paper has pretty good evidence suggesting a direct interaction between SET-26 and HCF-1.

Thank you for your comment. We have cited this paper accordingly.

Figure 6 Here you've focused on protein coding genes, but do you see any changes globally? For example, if you call peaks in your WT and set-24 experiments, do they mostly overlap? Given the subtle increase in H3K4me3 signal in set-24 mutants, perhaps you see an increase in peaks at non-protein coding regions of the genome too?

Thank you for your comments. The H3K4me3 peaks identified in WT and *set-24* largely overlap (**Fig. S10a**). We observed that approximately 10% of H3K4me3 peaks are increased in the *set-24* mutant, indicating that these changes are not global (**Fig. S10b**). Although the increased H3K4me3 peaks in *set-24* mutants can occur in non-protein-coding regions, such as ncRNAs, the majority (over 60%) are associated with protein-coding genes (**Fig. S10c**). Therefore, we focused our subsequent analyses on protein-coding genes. (see lines **350-353**)

It is difficult to understand how the analysis for determining H3K4me3 enriched genes has been carried out. From reading the methods (lines 685-686) and figure legend, it seems that the H3K4me3 enriched genes have been determined based on enrichment over input in WT (i.e. the genes that are targeted by this mark in WT), and then this subset is shown to be further enriched in H3K4me3 in set-24 mutants. However, the text in lines

267-269 suggests that the H3K4me enriched genes (note typo in fig 6a top panel) enrichment was determined in set-24 mutants. Can this please be clarified?

We apologize for the confusion. In wild-type worms, H3K4me3 accumulates at the TSS of genes. When *set-24* is mutated, a subset of these genes, those already marked with H3K4me3, show further increases in H3K4me3 levels. We refer to this subset as the H3K4me3-enriched genes. Accordingly, we revised lines 267–269 (in the previous version) to read: **“Importantly, the increase in H3K4me3 was not genome-wide but was restricted to genes with TSSs already marked by H3K4me3 in the wild-types (Fig. 7b). We identified a subset of genes with significantly elevated H3K4me3 levels in set-24(*syb7014*) mutants, hereafter referred to as H3K4me3-enriched genes (Figs. 7a-c).”** The revised sentences are now in the lines **355-359**.

Thank you for pointing out the typo, we have also corrected it in the **Fig. 7a**.

The ChIP/mRNAseq/sRNA is performed in whole worms, while SET-24 is expressed in the germline, thus presumably diluting/obscuring the signal. Could you perform these experiments on dissected germlines instead? Or could you take an alternative approach and look for changes in H3K4me3 in the germline by IF?

We agree with the reviewer that the ideal experiments would involve dissected germlines, as using whole worms may dilute the signal, particularly for H3K4me3 ChIP. However, performing these sequencing experiments on dissected germlines was not feasible due to insufficient material. In any case, by using whole-worm material, we are likely underestimating the potentially greater differences present in the germline.

To further validate SET-24’s role in regulating H3K4me3, we measured total H3 levels both globally and at SET-24–regulated H3K4me3-enriched genes in the ChIP-seq. H3 levels were unchanged between strains, indicating that the increase in H3K4me3 in *set-24* mutants represents a genuine modification change rather than altered H3 abundance (**Figs. S10b and S10e**). Comparison with a published germline expression dataset revealed that approximately 90% of the 757 *set-24*–dependent, H3K4me3-enriched genes are expressed in the germline⁴ (**Fig. S10f**), further supporting a role for SET-24 in regulating H3K4me3 in the germline, and suggesting that the majority of the differences emerge from germline signal. (please see lines **359-367**)

Typo in 6f – I think the middle panel should read CPM (mRNA) and the bottom panel should be CPM (22G-RNA)

Thank you for pointing this out. We have corrected the **Fig. 7g** accordingly.

Reviewer #2 (Remarks to the Author):

In this manuscript the authors describe a function for SET-24 in the maintenance of transgenerational epigenetic inheritance (TEI) through the balance of DNA methylation and small RNA production. They emphasize two major findings that distinguish SET-24 from previously described TEI factors: 1) SET-24 is a methyltransferase that lacks both the canonical catalytic motif and the canonical DNA-binding PHD motif and 2) SET-24 is specifically required for maintenance but not for establishment of TEI responses. Major revisions are needed, specifically to strengthen the link between SET-24 and its actions on chromatin/DNA and to clarify how SET-24's impacts on chromatin/small RNA contribute to the Mrt TEI phenotype.

Thank you for your comments, and we apologize for not clearly highlighting our findings. SET-24 contributes to the maintenance of RNAi-induced transgenerational inheritance and regulates small RNA (22G-RNA) population and **the histone mark H3K4me3, rather than DNA methylation**, which is generally thought to be absent in *C. elegans*⁵. Unlike members of the scSET3 subfamily (e.g., HsMLL5, UpSET, CeSET-9/SET-26, ScSET3), SET-24 lacks the PHD domain. To our knowledge, **the PHD domains in this subfamily interact with histone modifications, particularly H3K4me3, rather than DNA**⁶⁻⁹.

Major points:

Unfortunately, how SET-24 interacts with DNA seems murky at this point, a major missing link. Not only would an understanding of this interaction bridge the TEI phenotype to the downstream H3K4me3 and small RNAs mechanisms, it also would provide a new perspective concerning how a catalytically inactive methyltransferase can act to influence chromatin state. The authors themselves propose a likely scenario –the SPK (MYB-like) domains participate in ds-telomeric DNA binding. The authors should (1) delete/mutate the SPK domains site-specifically and assay TEI; (2) delete/mutate the SPK domains site-specifically and check for SET-24 mislocalization from DNA regions using their fluorescently-tagged SET-24 strain; and/or (3) add to their AlphaFold data and look at potential DNA-binding conformations of SET-24.

Thank you for your comments. TEBP-1 and TEBP-2 each contain three MYB domains, but only one directly interacts with DNA despite their structural similarity^{10,11}. The SPK domains of SET-24 share high structural similarity with these MYB domains, suggesting they might also bind DNA (Fig. S2c). To address this, we established a band shift assay using a DNA-binding protein (the tandem helix-turn-helix domains of the Mos1 transposase) and its known DNA target, as positive control^{2,3}. In this case, a clear DNA band shift is observed (**Revision Figs. 1a and 1b**). We then expressed and purified SPK domains *in vitro* (**Revision Figs. 1c and 1d**). As the potential direct DNA targets of the SPK domains remain unknown, we tested the ability of the SPK domains to bind to several

H3K4me3 SET-24 targets (as defined in Fig. 7a and Supplementary table 4; genomic regions amplified by PCR), or to a DNA library containing random sequences of the same size, but no band shift was observed (**Revision Figs. 1e-1g**). This suggests that the SPK domains in SET-24 may not bind DNA directly. Alternatively, the DNA sequences tested may not be direct targets.

Revision Fig. 1 (for the reviewer only) **a** Coomassie blue staining of purified GST and GST-MOS1-HTH, the two HTH domains of the transposase Mos1 that have known DNA-binding ability. **b** Electrophoretic mobility shift assay (EMSA) demonstrating high-affinity binding of the isolated Mos1 transposase double-HTH domain to its cognate terminal inverted repeat DNA sequence. **c** Schematic diagram of the GST-tagged constructs with SPK domains of SET-24. **d** Coomassie

blue staining of the purified GST-tagged SPK domains. **e** EMSA of the SPK domains with a mixture of 200 bp DNA fragments from H3K4me3-enriched genes. **f** Schematic of a double-stranded nucleic acid pool containing random sequences of fixed length. The single-stranded DNA library, containing primer binding regions at the 5' and 3' ends, was synthesized by IDT. The double-stranded DNA pool was then generated by PCR amplification using primers complementary to the binding regions and using the single-stranded DNA library as template. **g** EMSA of the SPK domains with the random double-stranded DNA pool. The DNA-binding protein MOS1-HTH did not show a band shift with the double-stranded DNA pool, likely due to the insufficient representation of the target sequence in the random-sequence DNA pool.

We also used AlphaFold3 to model potential interactions between the SPK domains of SET-24 and telomeric repeats, which are typically recognized by related telomere-binding MYB domains. However, no reliable interactions were observed, possibly due to the DNA-binding sequence of MYB domains is not a good proxy for the SPK domains.

In conclusion, given the lack of clear evidence for a potential DNA-binding motif in SET-24, we are currently unable to confirm the hypothesis that SPK domains bind DNA in a MYB-like manner. Developing an experimental system to systematically investigate DNA binding by SPK, an uncharacterized protein fold with a yet unknown potential DNA target, falls beyond the timeline of this revision and the scope of the paper.

We agree that deleting or mutating the SPK domains and examining the resulting Mrt/TEI phenotypes and SET-24 localization would be an effective approach. Unfortunately, due to time and material constraints, our attempts to generate these mutants have so far been unsuccessful. As we can't predict the time required to generate these strains, we plan to pursue this line of investigation in future research.

Related to the above issue, SET-24 might be associating with HCF-1 to bind DNA (even though they are functional opposites phenotypically for TEI). Clarity about how SET-24 and HCF-1 affect each other besides being binding partners should be addressed. Human MLL5 and HCFC1 have previously been shown to be interacting partners, and this fact should be stated explicitly. The relevant citation, #78 is in the References but seems to be describing something else in the text. set-24 (lf) and hcf-1 (lf) do not affect each other's localization as visualized. We recognize that the current reagents for localizing both SET-24 and HCF-1 are both GFP-based, but the 5th paragraph of the main text is subtitled "The SET-24 interactor, HCF-1 ..." and it is important to bolster this claim by one or more of the following experiments: (1) Imaging both SET-24 and HCF-1 at higher resolution to check for co-localization, which should be easily done, perhaps by imaging the HCF-1::GFP strain with an antibody label (using a separate non-GFP green color) against the FLAG on the SET-24::3xFLAG and with DAPI to visualize the DNA. (2) SET-24 and HCF-1 are proposed to be interacting partners, yet they are shown to act in opposing directions for the TEI phenotype. To clarify how this interaction is regulated, a double

mutant should be constructed and tested for TEI or signs of epistasis in the pathway. (3) The authors might also check for the previously identified (in human) HCF-1-binding motif (HBM) site in SET-24 as mentioned in reference #78 and perhaps give more insight into how that motif fits in their AlphaFold models and implications for how the potential HCF-1 binding site interacts structurally with the SET-24 SPK domains that supposedly bind DNA (e.g., do they occlude each other sterically, does the structural orientation of the domains inform how the SET-24-HCF-1-DNA complex might work etc.). (4) An in vitro interaction assay, e.g., if purified SET-24 (using the FLAG IP) were found to bind HCF-1::GFP (GFP Ab IP) together with purified DNA that would provide evidence for a SET-24-HCF-1-DNA interaction.

Thank you for your comments. These comments help us a lot to improve our manuscript.

To address this comment, the revised version of the manuscript includes additional data (**Figs. 4b-4d, 5c-5i, 6a, 6e-6g, S6f, S6h, S7, and S8c-S8f**), which we will describe in detail below, and textual changes.

(1) Using confocal microscopy, we analysed the colocalization of SET-24::GFP and H2B::mCherry in live worms by calculating Pearson's colocalization coefficient in ImageJ. This analysis showed that SET-24::GFP partially colocalizes with H2B::mCherry in the nucleus (**Fig. 4b and 4d**). Conversely, the colocalization between SET-24::GFP and tagRFP::SYP-1, a marker of the synaptonemal complex, is weaker (**Fig. 4c and 4d**). With this resolution and by imaging live worms, we detected SET-24::GFP signal dispersed throughout germline nuclei, but also within puncta. The SET-24 puncta did not colocalize well with H2B::mCherry and tagRFP::SYP-1, suggesting that they are not associated with chromosomes (**Figs. 4b and 4c**).

Our original HCF-1::GFP strain from Siu Sylvia Lee's lab carried a 3×FLAG tag¹, making it impossible to perform IF staining for SET-24::3xFLAG, HCF-1::GFP::3xFLAG, and DAPI simultaneously. Instead, we generated a transgene expressing mKate2-tagged HCF-1 in the germline along with a BFP::H2B transgene to label the histone. We then examined colocalization of SET-24::GFP, HCF-1::mKate2, and BFP::H2B in the germline of live worms. As shown in **Figs. 5d and 5e**, SET-24 and HCF-1 colocalize strongly and both proteins are partially colocalized with H2B.

Figs. 5d and 5e

We next examined whether SET-24 and HCF-1 influence each other's localization using confocal microscopy and found that neither affected each other's localization (Fig. S6h).

Please see lines **223-231** and **277-283**.

- (2) Both *hcf-1* mutants exhibited reduced brood size at 20 °C and are nearly sterile at 25 °C within one or two generations, without producing “escapees”. We next examined the brood size of *set-24; hcf-1* double mutants and found that their progeny numbers were comparable to those of the *hcf-1* single mutants. (Fig. 6a)

In addition, we quantified the inheritance of RNAi silencing of a GFP reporter in *set-24; hcf-1* double mutants. We silenced heterozygous *+/set-24; +/hcf-1* worms carrying the GFP::H2B transgene by feeding *gfp* dsRNA and monitored silencing inheritance in their progeny (including a range of genotypes: wild-type, *set-24* or *hcf-1* single mutants, and *set-24;hcf-1* double mutants; see Fig. 6e). Consistent with prior observations, silencing was lost more quickly in *set-24* single mutants than in wild-type animals, whereas it was maintained for more generations in *hcf-1* single mutants (Figs. 6f and 6g). Importantly, *set-24; hcf-1* double mutants also maintained silencing longer than wild-type worms, similar to the *hcf-1* single mutants (Figs. 6f and 6g). This phenocopying was observed when experiments were repeated using a different genetic experimental design (Figs. S8c-S8f).

Figs. 6e-6g

We conclude that *set-24; hcf-1* double mutants phenocopy *hcf-1* single mutants in terms of fertility defects and RNAi inheritance. We propose that the opposite phenotypes of *set-24* and *hcf-1* mutants, and the dominance of the *hcf-1* mutant phenotype are due to the broader regulatory roles of HCF-1 in concert with multiple chromatin remodelling complexes. (Please see lines 321-335)

- (3) SET-24 protein **does not** encode a sequence identical to HCF-1 binding motif (HBM) in human. Now the interaction between human HCF1 and MLL5 is mentioned in the revised manuscript with the relevant reference. (Please see **Fig. S2a** and lines 285-287 and 464-467)
- (4) We used an *in vitro* pull-down assay to show that **SET-24 and HCF-1 physically interact (Figs. 5c and S6c)**. We went further and mapped the interactions to specific protein domain, using *in vitro* pull-down assays. These approaches revealed that residues 1–378 of HCF-1 (mainly containing the Kelch domain) directly interact with SET-24 (**Figs. 5f, 5h, and S7a**). All SET-24 truncations containing the N-terminal 1–239 region, which includes the SET domain and its flanking sequences, bound to the HCF-1 1-378 fragment (**Figs. 5g, 5i, S7b, and S7c**). Interestingly, the SET domain and some of its flanking sequences seem to be necessary for association with HCF-1 (**Figs. 5g, 5j, 5k, S7d, S7f**). Together, these data suggest the Kelch domain of HCF-1

directly interacts with the SET domain of SET-24, with the flanking sequences of the SET domain also contributing to the interaction. (Please see lines **268-270** and **285-301**)

Figs. 5c

In conclusion, we have provided additional microscopy, genetic, and biochemical evidence that further solidifies our claim that SET-24 and HCF-1 are bona fide interactors.

The H3K4me3 enrichment and any resultant effects on transcription in set-24(lf) mutants compared to wild-type worms appear mild. To support their claims that altered H3K4me3 levels and thus chromatin state is what SET-24 is affecting to regulate TEI, the authors should compare the levels of H3K4me3 and 22-G RNAs in the G3 set-24(lf) affected lines with those in the “escapee” wild-type-looking lines at G4 to see if H3K4me3 and 22-G RNA levels return to normal. This experiment might link proposed downstream mechanistic effectors of 22G-RNA levels to the organismal Mrt phenotype with good (generational) temporal specificity. Such a finding would be of considerable interest given the proposed adaptive biological functions of SET-24 discussed in the last paragraph of the paper.

In *set-24* mutants, we identified a subset of genes with elevated H3K4me3 levels, hereafter referred to as H3K4me3-enriched genes. H3 ChIP analyses in wild-type and *set-24* mutants revealed no global changes in H3 levels, and H3 occupancy at these enriched loci was not increased in *set-24* (**Figs. S10b and S10e**). Thus, although the elevation of H3K4me3 in *set-24* mutants is modest, the increased H3K4me3 signal likely reflects the regulatory effects of SET-24 rather than changes in histone abundance.

Apologies for the confusion. **We did not claim that the altered H3K4me3 levels and chromatin state in set-24 mutants are underlying the TEI defects.** SET-24 regulates H3K4me3 and thereby modulates chromatin state at a subset of genes, and it also influences RNAi-induced TEI. However, we agree that our current data are insufficient to conclude that H3K4me3 regulation by SET-24 underlies TEI maintenance.

We also fully agree that the “escapee” phenotype in the Mrt assay is very intriguing and could be key to uncover the dynamics of the deregulation in *set-24* mutants. We do note that the nature of the Mrt phenotype, with progressive decrease in fertility, makes it hard to collect biomass for experiments such as ChIP-sequencing, which require a substantial amount of biomass, simply not possible to obtain in a Mrt mutant. Likewise, it would be challenging to collect enough biomass for ChIP-sequencing experiments in the context of a TEI assay. This intrinsic limitation of the experimental design of these assays makes it intractable to profile small RNAs and chromatin in intermediate stages of the Mrt experiment where fertility is still in decline and populations dwindle. Even if such assays were possible in Mrt worms, another confounding factor would be the germline/somatic signal in whole worms. This is because Mrt worms display atrophied or severely reduced germlines, while P0/escaped worms have healthy germlines (see **Fig. 2c**). Therefore, the viable option we embarked on for the time being was to profile small RNAs in P0 and escaped populations.

We sequenced 22G-RNAs in wild-type and *set-24(syb7014)* mutant worms at 25 °C at both P0 and G12 generations (the latter expanded from a single G10 “escapee”; four replicates each condition). The 22G-RNA population at G12 (*set-24* vs. wild type) differs from that at P0 under 25 °C conditions (**Revision Figs. 2a-2c**), suggesting that 22G-RNA deregulation plays a role. The genes with deregulated 22G-RNA levels overlap partially between P0 and G12, in some cases even changing regulation (i.e. from down- to upregulated, or vice-versa; see **Revision Fig. 2c**). Thus, the small RNA deregulation associated with escape of Mrt phenotype is complex.

Because of the apparent complexity of the deregulation and stochasticity of the phenotype, additional comprehensive analyses are warranted, for example sequencing small RNAs on a substantially larger number of biological replicates of escapee populations. However, we believe these could be more effectively tackled in a dedicated study. Indeed, we are planning future studies to investigate how “escapees” arise in *set-24* mutants growing in high temperature conditions. Given the experimental caveats stated above, we plan to perturb the expression of a series of epigenetic factors involved in chromatin remodelling and small RNA pathways during a Mrt experiment. This approach may help identify which pathways, when disrupted, can block the occurrence of “escapees”.

Revision Fig. 2 (for the reviewer only) **a** Plots showing the proportion of small RNAs length distribution profiles, along with information on the first nucleotides in wild-type and *set-24(syb7014)* samples at 25°C P0 or G12 ("escapees" for *set-24* mutants), which were treated with RppH. **b** Venn diagrams showing overlap of targets of differentially expressed 22G-RNAs in *set-24(syb7014)* at 25°C P0 and G12 (fold change > 2 and 5% FDR), G12 worms were collected from the "escapees" lines. **c** Upset plot showing overlap of targets of upregulated- and downregulated 22G-RNAs in *set-24(syb7014)* at 25°C P0 and G12.

Relatedly, such an experiment could also be done with *HCF-1* mutants, which show opposing TEI effects, to see if the *H3K4me3* and 22G-RNA changes identified in *set-24(lf)* mutants are corresponding opposed in *hcf-1(lf)* mutants thus demonstrating mechanistic bidirectionality in tying the phenotypic observations of TEI to the molecular players.

hcf-1 mutants exhibit reduced brood size at 20 °C and become nearly sterile at 25 °C within one or two generations, without producing "escapees" (Fig. 6a). Consequently, analysing small RNAs or histone modifications at 25 °C is not feasible. *HCF-1* interacts

not only with SET-24 but also with multiple chromatin remodelling complexes, including the COMPASS complex and the H3K4me3 methyltransferase SET-2. Therefore, while alterations in H3K4me3 levels are expected in *hcf-1* mutants, the function of HCF-1 is likely broader and not limited to its interaction with SET-24.

As suggested, we sequenced 22G-RNAs in *hcf-1* mutants at 20 °C (Figs. S11b, S11f, and S11g). In both *set-24(syb7014)* and *hcf-1(pk924)* mutants, 22G-RNA populations were altered, with more 22G-RNAs upregulated than downregulated (Fig. S11b). A subset of 22G-RNAs was regulated by both *hcf-1* and *set-24*; however, most were regulated by only one gene, particularly *hcf-1*, despite the direct interaction between HCF-1 and SET-24 and their shared contribution to TEI (Fig. S11f). This likely reflects HCF-1's broader interaction network with multiple chromatin remodelling complexes, which may also affect small RNA regulation. Within the subgroup of 22G-RNAs regulated by both *hcf-1* and *set-24*, some (94/269) were affected in opposite directions, but the majority (175/269) were not (Fig. S11g). These findings suggest that the opposing phenotypes of *hcf-1* and *set-24* cannot be explained solely by shared 22G-RNA targets, but instead may involve additional mechanisms, such as interactions between HCF-1 and other factors. We plan to further investigate how HCF-1 inhibits the maintenance of silencing in future studies. (please see lines 396-403)

Fig. S11b

Figs. S11f and S11g

Minor points:

The authors should explain why the wild-type responses for depressed GFP-RNAi differ quite dramatically between Fig 3b (<10-20% at G7) and Fig 5d (close to 100% at G7)?

Thank you for your comments. We apologize for not making it clear.

For Fig. 3b, germline nuclear GFP brightness was scored by visual inspection as 1 (“bright”) or 0 (“dark”), with dim GFP expression also classified as 0. For all other figures containing GFP RNAi inheritance assays, including Fig. 5d from the previous submission (now Figs. 6c, 6d, 6f, 6g, S8a, and S8c–S8f), nuclear GFP brightness was assessed visually by a different team member, with “bright” scored as 1, “dark” scored as 0, and “dim” scored as 0.5. This scoring of “dim” may contribute to differences observed in wild-type samples across figures. This clarification has been added to the Methods section, see lines **647-652**. We note that the phenotype of the mutants relative to wild-type controls is consistent across experiments, despite distinct scoring strategies, highlighting the robustness of our results.

In addition, the duration of inheritance is influenced by factors such as the strength of the initial RNAi trigger, the bacterial strain used for feeding, and other unknown environmental variables. The GFP RNAi inheritance assays for wild-type animals shown in different figures were conducted at different times. Although we attempted to maintain consistent experimental conditions, the baseline of the RNAi response often differs. However, the wild-type controls we include in each experiment allow for adequate, direct, and internal comparison with the mutant strains in the same experiment. While the baseline differs between experiments, the mutant effects versus wild-type controls are consistent across all the experiments.

Altogether, we are confident that our conclusions remain robust.

The authors have presented a comprehensive characterization of the phenotypic TEI consequences of the loss of SET-24. The authors make a case of the striking G3 vs G4 Mrt difference, attributing it to “escapee” takeover of the population. There should be a more detailed population breakdown or description of this rather sharp transition reversal. If one were to pick Mrt-displaying adult worms with degenerated germlines in G3 and assay their G4 progeny what would be the outcome? Would all G4 progeny be sterile/non-viable? As it stands the increase in animals having normal germlines past G4 might be due to reversion or compensation mechanism being activated rather than “escapee” takeover and these scenarios could be distinguished.

Thank you for your comments, and we apologize for the confusion. To clarify, we have included a schematic (**Fig. S3b**) illustrating how defective germlines were scored. For each generation, more than 30 worms were examined, sampled evenly from independent lines rather than from chunked plates. In Fig. 2c, by G3, most plates were nearly sterile or displayed sterility tendency, and consequently, the majority of worms sampled exhibited defective germlines. From G4 onward, however, many plates contained escapee lines, and most worms sampled displayed normal germlines. **Importantly, our observations indicate that Mrt-phenotype worms (such as those with defective or absent germlines) from G3 retained their phenotype when propagated to G4 and beyond, ultimately becoming sterile.**

Fig. S3b

SET-24's germline-specific expression was previously established (albeit only for L1 stages) by Engert et al. (2018) (doi:10.1371/journal.pgen.1007295). Given the emphasis on SET-24 expression in this submitted manuscript, this fact should be explicitly stated and this earlier paper should be cited.

Thank you for your comments. We have cited the paper accordingly in **lines 202-203**.

Are any of the identified WAGO and Mutator genes identified in this paper possible effectors of the Mrt phenotype? Have they appeared in screens / sequencing from studies involving the Mrt phenotype?

Thank you for your comments. From our small RNA sequencing data (**Figs. 7e and S11d**), we found that *set-24* regulates the population of 22G-RNAs, and the 22G-RNAs regulated

by *set-24* largely overlap with WAGO- and Mutator-class target genes, which have previously been reported to be regulated by WAGOs and Mutator factors¹²⁻¹⁶. To be clear, these target genes are not WAGO or Mutator genes themselves.

We generated lists of overlapping genes between *set-24*-regulated 22G-RNA targets and WAGO- or Mutator-class 22G-RNA targets (**Supplementary Table 7**). We then manually examined the observed RNAi and allele phenotypes associated with these genes using Wormbase Simplemine (<https://wormbase.org/tools/mine/simplemine.cgi>), but we did not identify any factors involved in Mrt or TEI regulation. In conclusion, while we cannot fully exclude that SET-24 regulates a Mrt effector, our data does not provide any supporting evidence for this.

Reviewer #3 (Remarks to the Author):

In this manuscript from Zeng et al, the authors examined a SET-domain containing protein (SET-24) that plays roles in germline immortality and TEI in C. elegans according to their previous published work. The SET domain is typically the catalytic domain of histone methyltransferase. However, the authors showed that SET-24 shares sequence similarity with the scSET3 superfamily members that are catalytic inactive. Therefore, whether SET-24 can methylate histone is unclear. Interestingly, SET-24 is expressed in germline lineage of the C.elegans, and is required for germline immortality and heritable RNAi, suggesting it plays important roles in epigenetic inheritance. Using IP-Mass Spec and yeast two-hybrid assay, the authors show that SET-24 interact with HCF-1, a key component of the COMPASS chromatin remodeling complex that regulate H3K4me3. The author then examine the role of SET-24 in regulating H3K4me3, mRNA and small RNA expression. In set-24 mutant, a mild increase of H3K4me3 is observed at a subset of transcription start sites(TSS), but few corresponding mRNA changes were observed. In contrast, more small RNA changes are seen at these TSS, and these small RNAs belong to the WAGO and Mutator regulated genes.

Overall, the manuscript provides strong evidence that SET-24 is required for germline immortality and TEI, offering valuable insight into the field of epigenetic inheritance. However, the modest H3K4me3 changes at a subset of TSS in the set-24 mutant also raise the questions about whether these loci are direct targets of SET-24/HCF-1 (COMPASS complex). The genome-wide analysis (ChIP-seq, mRNA-seq, small RNA-esq) lacks information regarding biological replicates and the allele used and requires major revision.

Thank you for your comments. We added all the information about biological replicates and the allele information we are using.

Specific comments:

1. The structure-based phylogenetic analysis supports that key residues involving in catalytic activity in the SET domain are missing in SET-24, SET-9, and SET-26. However, SET-26 has also been reported to have H3K9 HMT activity in vitro (PMID: 24685137). It remains possible that these SET domains utilize alternative residues for catalytic function.

Thank you for your comments. To experimentally examine whether SET-24 is catalytically active, we tested the methyltransferase activity of SET-24 on histones *in vitro* using a well-established non-radioactive methyltransferase assay (Fig. 1d). We purified MBP-tagged full-length *C. elegans* SET-24 (CeSET-24) and *S. cerevisiae* SET2 (ScSET2), an active H3K36me3 methyltransferase, as a positive control^{17,18} (Fig. S1a). While the assay

confirmed a robust methyltransferase activity of ScSET2, CeSET-24 did not exhibit activity on any of the four core histones in our assay (**Fig. 1e**). We conclude that SET-24 lacks detectable histone methyltransferase activity. (please see lines **114-120**).

Figs. 1d and 1e

For the other SET factors that lack the canonical residues required for the methyltransferase activity, as mentioned in the manuscript, we describe them as “likely catalytically inactive”. (Please see lines **120-123**)

2. The SET-domain of SET-24 appears to end before the 144 a.a. (Fig. 1C, Fig. S1). The set-24 *mf123/mj617* allele is truncated after the 188 a.a. (Fig. 1D). It is not clear if the SET-domain remained in this mutant.

Thank you for pointing this out, we apologize for this lapse.

Indeed, the *mf123/syb7014* allele carries a premature stop codon located after the SET domain. We have updated the schematic in **Fig. S3a** accordingly. Although the SET domain is still present, we note that it is very likely that it is not expressed, as *set-24(mf123)* and *set-24(syb7014)* phenocopy the null allele *set-24(mj617)*.

3. Fig. 4C: SET-24 form puncta in the embryo P-lineage. Some SET-24 puncta appear to be cytoplasmic and is enriched in PGL-3 foci.

That is a good point. We have updated the text accordingly in lines **220-221**, “In embryos, SET-24::GFP was solely expressed in germline-lineage cells (Fig. 4a).”

4. Line 217-222, Fig. S6: ChIP might still work even without clear DNA enrichment. This could be due to higher background (since SET-24 only expressed in a subset of germline line cells) or fewer binding sites. Sequencing these samples might still reveal the binding sites.

This is a great recommendation, thank you. We agree that this would be a valid approach to identify SET-24's binding sites. We had sequenced the SET-24 ChIP products, nevertheless none or extremely low throughputs were obtained. We conclude that either unknown technical aspects affected the pulldown or SET-24 does not stably associate with chromatin.

5. The detection of HCF-1 in both the IP-MS and Y2H supports an interaction between SET-24 and HCF-1. How many biological replicas were performed for these experiments? Were other COMPASS complex components identified in the IP-MS? Including a co-IP validation of SET-24 and HCF-1 would further strengthen the conclusion.

Thank you for your comments. The revised manuscript includes additional GST-pulldown assay data strengthening our conclusion (**Figs. 5c, 5f-5k, S6c, and S7**, please see lines **268-270** and **285-301**).

Please note that HCF-1 was identified in both approaches, which significantly strengthens the credibility of the result. The Y2H experiment was performed by HYBRiGENiCS SERVICES (<https://www.hybrigenics-services.com>) under their standard quality controls. The IP-MS was done through label-free quantification, using IPs from **four** biological replicates of wild-type and SET-24::3xFLAG strains. This approach was used extensively in similar IPs of FLAG-tagged proteins from *C. elegans* samples, yielding reproducible results^{10,12,19}. The relevant information has been included in the corresponding methods (Please see the lines **753** and **761-774**). In both IP-MS and Y2H screenings, we did not detect other COMPASS components.

We further performed *in vitro* pull-down assays and confirmed that SET-24 directly interacts with HCF-1 *in vitro* (**Figs. 5c and S6c**). In addition, we identified specific regions of SET-24 and HCF-1 that mediate their interaction (**Figs. 5f-5k and S7**). Altogether, **the IP-MS, Y2H, and *in vitro* pull-down data clearly indicate that SET-24 and HCF-1 interact directly.**

Fig. 5c

6. Line 238, Fig5: Is the *hcf-1(ok559)* mutant a null mutant? More information on this allele is necessary to interpret HCF-1's role in the fertility assay and heritable RNAi assay.

Thank you for your comment. *hcf-1(ok559)* is not a null mutant but carries an in-frame deletion in the Kelch domain and the *hcf-1(ok559)* mutants could still express truncated HCF-1 *in vivo*²⁰. We have included the allele information of *hcf-1* in **Fig. S6g**. (Also see the lines **305-307**)

To further validate the phenotypes observed in *hcf-1(ok559)*, we used an alternative *hcf-1* allele, *hcf-1(pk924)*, to perform the Mrt and TEI assays. As shown in **Figs. 6a-d and S8b**, both *hcf-1* alleles exhibit very similar phenotypes, indicating that the Kelch domain of HCF-1 plays an essential role in viability and TEI maintenance. In fact, the revised manuscript includes new data with the identification of the Kelch domain as the domain responsible for interaction with SET-24 (**Figs. 5h and S7a**). Thus, the in-frame deletion of the Kelch domain is a relevant model to study the roles of HCF-1 in relation to SET-24 regulation.

7. Which of the two *set-24* mutants were used for the H3K4me3 ChIP-seq? Do both *set-24* alleles exhibit the same changes in H3K4me3 levels? How many biological replicates were included and how are they? The observed H3K4me3 increase at a subset of TSS of *set-24* mutant is modest (<20%, based on Fig 6b), raise the question of whether these TSS sites are the direct or indirect target of the SET-24/HCF-1 (COMPASS complex).

Thank you for your comments! We used ***set-24(syb7014)*** for the H3K4me3 ChIP-seq and prepared three replicates each for wild type and *set-24(syb7014)* for sequencing. However, one of the H3K4me3 ChIP fails in the library preparation. Therefore, in **Fig. 7**, there are **two** replicates for *set-24(syb7014)* and *wild-type*. This information has been added to the corresponding figures (Please see the **figure legend of Figs. 7a and S10b**). We did not perform H3K4me3 ChIP-seq for both *set-24* alleles, as they exhibit very similar phenotypes.

Since the direct or indirect DNA targets of SET-24, as well as the DNA targets of germline HCF-1, remain unknown, we were unable to compare the H3K4me3-enriched genes with SET-24 or germline HCF-1 targets. We acknowledge this as a limitation of our study and hope to address it in future research.

8. Similarly, the mRNA-seq and small RNA-seq analyses lack details about which *set-24* allele was used and how many biological replicates were performed.

Apologies for the omission. We have now stated this information clearly in the figure legends. (Please see the **figure legend of Figs. 7d, 7e, S11a, and S11c**)

For the mRNA-seq, we used the **set-24(syb7014)** allele with **four** replicates for each strain. For the small RNA-seq, we used **both set-24(syb7014) and set-24(mj617)** alleles, with **four** replicates for each strain. Each mutant had its own corresponding wild-type controls, collected at the same time. For *set-24(syb7014)* and its wild-type controls, the samples were the same as those used for mRNA-seq. Although the *syb7014* allele showed stronger deregulation of 22G-RNAs, the profiles of the two alleles correlated well (**Figs. S11c**) and showed similar overlaps with WAGO and Mutator targets (**Figs. 7e and S11d**).

In our first submission, we used *set-24(mj617)* for the small RNA analysis. To maintain consistency across the H3K4me3 ChIP-seq, mRNA-seq, and small RNA-seq datasets, in the revised manuscript we placed the *mj617* small RNA data in the supplement **Figs S11d**, and the *syb7014* allele data in **Figs. 7f and 7g**. (Also see the lines **387-389**)

9. Since SET-24 is germline-specific, how many of the 757 genes showing SET-24-dependent H3K4me3 are also germline expressed? This analysis could help support a direct role for SET-24 in regulating H3K4me3 via HCF-1. Additionally, do the genes with altered small RNA levels in the set-24 mutant also fall into the category of germline-expressed genes?

Thank you for your comments. To assess the germline expression of the 757 *set-24*-dependent H3K4me3-enriched genes, we intersected them with a published dataset of germline-expressed genes⁴. Our analysis revealed about 90% of *set-24(syb7014)*-dependent H3K4me3-enriched genes and approximately 60% of *set-24(syb7014)*-regulated 22G-RNA targets are expressed in the germline (**Figs. S10f and S11e**). This indicates that the deregulation of H3K4me3 and 22G-RNAs in *set-24* mutants has a strong germline signal. (please see the lines **364-367** and **396-399**)

Figs. S10f and S11e

Minor comments:

Line 178: “transcript sand” should be “transcripts and”.

Figure 6f: Can not find track labels for sRNA-seq or mRNA-seq.

Thank you for pointing these out. We updated the texts and figures accordingly. (Please see the line **185**).

Reviewer #4 (Remarks to the Author):

Thank you!

Reference

1. Emerson, F.J. et al. The chromatin factors SET-26 and HCF-1 oppose the histone deacetylase HDA-1 in longevity and gene regulation in *C. elegans*. *Nat Commun* **15**, 2320 (2024).
2. Almeida, M.V. et al. Transposable Elements Drive Regulatory and Functional Innovation of F-box Genes. *Mol Biol Evol* **42**(2025).
3. Richardson, J.M., Colloms, S.D., Finnegan, D.J. & Walkinshaw, M.D. Molecular architecture of the Mos1 paired-end complex: the structural basis of DNA transposition in a eukaryote. *Cell* **138**, 1096-108 (2009).
4. Ortiz, M.A., Noble, D., Sorokin, E.P. & Kimble, J. A new dataset of spermatogenic vs. oogenic transcriptomes in the nematode *Caenorhabditis elegans*. *G3 (Bethesda)* **4**, 1765-72 (2014).
5. Wenzel, D., Palladino, F. & Jedrusik-Bode, M. Epigenetics in *C. elegans*: facts and challenges. *Genesis* **49**, 647-61 (2011).
6. Tran, K. & Green, E.M. SET domains and stress: uncovering new functions for yeast Set4. *Curr Genet* **65**, 643-648 (2019).
7. Ali, M. et al. Molecular basis for chromatin binding and regulation of MLL5. *Proc Natl Acad Sci U S A* **110**, 11296-301 (2013).
8. Wang, W. et al. SET-9 and SET-26 are H3K4me3 readers and play critical roles in germline development and longevity. *Elife* **7**(2018).
9. Carvalho, C.A. et al. SUMO-mediated regulation of H3K4me3 reader SET-26 controls germline development in *C. elegans*. *PLoS Biol* **23**, e3002980 (2025).
10. Dietz, S. et al. The double-stranded DNA-binding proteins TEBP-1 and TEBP-2 form a telomeric complex with POT-1. *Nature Communications* **12**(2021).
11. Padmanaban, S. et al. *Caenorhabditis elegans* telomere-binding proteins TEBP-1 and TEBP-2 adapt the Myb module to dimerize and bind telomeric DNA. *Proc Natl Acad Sci U S A* **121**, e2316651121 (2024).
12. Almeida, M.V. et al. GTSF-1 is required for formation of a functional RNA-dependent RNA Polymerase complex in *Caenorhabditis elegans*. *Embo j* **37**(2018).
13. Claycomb, J.M. et al. The Argonaute CSR-1 and its 22G-RNA cofactors are required for holocentric chromosome segregation. *Cell* **139**, 123-34 (2009).

14. Zhou, X. et al. Nuclear RNAi contributes to the silencing of off-target genes and repetitive sequences in *Caenorhabditis elegans*. *Genetics* **197**, 121-32 (2014).
15. Phillips, C.M. et al. MUT-14 and SMUT-1 DEAD box RNA helicases have overlapping roles in germline RNAi and endogenous siRNA formation. *Curr Biol* **24**, 839-44 (2014).
16. Gu, S.G. et al. Amplification of siRNA in *Caenorhabditis elegans* generates a transgenerational sequence-targeted histone H3 lysine 9 methylation footprint. *Nat Genet* **44**, 157-64 (2012).
17. Du, H.-N., Fingerman, I.M. & Briggs, S.D. Histone H3 K36 methylation is mediated by a trans-histone methylation pathway involving an interaction between Set2 and histone H4. *Genes & Development* **22**, 2786-2798 (2008).
18. Strahl, B.D. et al. Set2 Is a Nucleosomal Histone H3-Selective Methyltransferase That Mediates Transcriptional Repression. *Molecular and Cellular Biology* **22**, 1298-1306 (2003).
19. Marnik, E.A. et al. The *Caenorhabditis elegans* TDRD5/7-like protein, LOTR-1, interacts with the helicase ZNFX-1 to balance epigenetic signals in the germline. *PLOS Genetics* **18**(2022).
20. Lee, S. et al. Epigenetic regulation of histone H3 serine 10 phosphorylation status by HCF-1 proteins in *C. elegans* and mammalian cells. *PLoS One* **2**, e1213 (2007).

Response to REVIEWER COMMENTS

Reviewer #1 (Remarks to the Author):

I would like to commend the authors on the large amount of extra work that they have performed and included in this revised manuscript. Below are some comments on the additions, in the context of what I wrote in my previous review.

Given the increased focus on the opposite effect of HCF-1 and SET-24, is the title still appropriate? It does make it sound as though they act in the same way, when actually they have opposite effects on epigenetic memory.

I'm also still not convinced that the authors have really nailed the mechanism by which SET-24 acts to regulate TEI.

Thank you for reviewing our manuscript again. Although the loss of SET-24 and HCF-1 results in opposing phenotypes in heritable RNAi, both factors function in the maintenance of this process. Therefore, we believe the current title is concise and meaningful.

In this work, we demonstrate that the non-catalytic SET-domain-containing protein SET-24 is essential for germline integrity and for the maintenance of heritable RNAi across generations. SET-24 and its interactor HCF-1 are both required for sustaining heritable RNAi, despite their depletion producing opposing phenotypes. Moreover, SET-24 is crucial for maintaining the 22G-RNA population and proper chromatin state. We acknowledge that further work is needed to elucidate the precise mechanisms by which SET-24 regulates TEI. Nonetheless, our study provides solid evidence regarding SET-24's functions, localization, and interaction with HCF-1. We believe our findings offer broader insights into the roles of SET factors and provide new perspectives for future research on transgenerational epigenetic inheritance.

My previous main points

1. How does SET-24 interact with HCF-1? The authors have added a lot of new data to the manuscript to address this point, and have shown that the Kelch domain of HCF-1 interacts with the SET domain (and its immediate flanking regions) of SET-24. They furthermore show that loss of one protein does not affect the localization of the other, using fluorescent reporter strains.

2. Do they see changes in histone acetylation in set-24 mutants? They did not see any global changes to histone acetylation (although it's worth noting that they also didn't see global changes to H3K4me3 by western blot).

3. Do the SPK domains of H3K4me3 bind DNA as they speculate? The authors recombinantly express the SPK domain and tested them in a band shift assay, but did not see any evidence of DNA binding. I appreciate the effort, and acknowledge that absence of evidence is not evidence of absence, so leaving the comment in the Discussion (lines 445-446) is probably OK. I still do think that answering this question would be informative for the mechanism.

4. What is the SET domain doing? They showed that the SET domain seems to be required for stability of SET-24, as mutations in the SET domain in worms could no

longer be visualized with the GFP reporter, and protein could no longer be detected by western blot (although mRNA could). This is a good advance to the paper.

5. What are the phenotypes of *hcf-1::set-24* double mutants? They show that in all cases tested (not just the epigenetic inheritance assay), double mutants phenocopy *hcf-1* mutants, and propose that this is due to HCF-1 having broader roles with other chromatin remodelling complexes. I think these results really highlight the fact that although SET-24 and HCF-1 seem to be interacting, it's really not clear how this fits into the broader context of TEI.

6. What is going on with the escapees? I suggested various 'omics analyses could be performed. The authors did some small RNA seq on one population of escapees and showed dysregulation of small RNAs – not included in the manuscript. They also suggest that other experiments are ongoing and will form the basis of a later manuscript. I'm left a bit unsatisfied, but I guess this is OK on balance, give the amount of other data that they've added.

7. Does SET-24 colocalise with chromatin? The authors have addressed this well by doing colocalization analysis with H2B::mCherry and tagRFP::SYP-1, and show that SET-24 only partially localises with chromatin. Worth pointing out is the apparent expression of SET-24 in granules (overlapping partially with PGL-3) in the P3 or P4 cells. This is non-nuclear (although still germline)! The authors should comment on this in the text. It might be meaningful for the action of SET-24 in the context of small RNA regulation.

Thank you for your support of our manuscript, and we appreciate your understanding regarding the limitations of some experiments. For point 7, we have addressed this comment at lines 223-224. "In embryos, SET-24::GFP was solely expressed in germline-lineage cells, localized mainly to the nucleus, but also showing a partial overlap with perinuclear PGL-3".

Other comments

1. The recombinant proteins used for the in vitro methyltransferase assays are not at all well purified. There are many contaminants in the Coomassie gel (S1a). Can the authors provide more information to prove that that have indeed successfully expressed MBP-CeSET-24 and that it is folded correctly? If it isn't folded correctly (or expressed well) this would explain the negative result.

Thank you for your comment. Below, we summarize several lines of evidence showing that MBP-SET-24 was indeed expressed and that it lacks detectable activity in our methyltransferase assay.

First, we performed a Western blot using an MBP antibody on the protein sample used for the methyltransferase assay. A clear band corresponding to the expected molecular weight of full-length MBP-SET-24 was detected. This band migrates at the same molecular weight as the red star marked in the Coomassie gel. The bands below the full-length MBP-SET-24 band are mainly degradation products, which is common for large

proteins expressed recombinantly in *E. coli* to generate such breakdown products, as is also observed for MBP-ScSET2. (Please see supplementary Fig. 1a)

Supplementary Fig. 1a

Second, N-terminal tag does not appear to impair SET-24 function, as it is capable of interacting with HCF-1 (Figs. 5c and 5h, Supplementary Figs. 7a-7c).

Lastly, because the SET domain of SET-24 is located in the N-terminal region, the degradation products above ~66 kDa detected by the MBP antibody should contain the SET domain. If either the full-length MBP-SET-24 or any degradation product containing the SET domain were enzymatically active, we would expect to detect histone methylation in our assay. However, no activity was observed, indicating that SET-24 lacks detectable histone methyltransferase activity under our assay conditions.

Reviewer #2 (Remarks to the Author):

I am satisfied with the alterations the authors have made to the paper and accept their explanations for the limitations in certain proposed experiments as either technically or otherwise beyond the scope of the paper. I appreciate the increased clarity about the characterization of SET-24, its strength of interaction with HCF-1 and analysis of the HCF-1 TEI phenotype.

Thank you for reviewing our manuscript again. We appreciate your support of our work.

I have the following specific additional minor comments:

1) It seems the proposed model is that SET-24 and HCF-1 are acting antagonistically

(and coordinately) in a balance to maintain proper (wild-type) heritable RNAi generational duration. In lines 422-423, for example, it would be both clearer and more accurate to explicitly spell out the molecular-phenotypic description with appropriate directionality rather than just saying “required for heritable RNAi,” which could easily be read as HCF-1’s acting in the same direction as SET-24.

Thank you for your comment. We have revised the sentences as “SET-24 is essential for maintaining germline integrity and heritable RNAi across generations. Intriguingly, SET-24 and its interactor HCF-1 are both required for the maintenance of heritable RNAi, yet their depletion produced opposing phenotypes.” Please see lines 429-432.

2) SET-24 and HCF-1 act antagonistically in contributing to the TEI phenotype, and *hcf-1* single mutants and *hcf-1*; *set-24* double mutants are equivalent in phenotype, i.e. *hcf-1* shows “epistasis” not “dominance” to *set-24* (line 335). The authors might make explicit in that context the statement about *hcf-1*’s being downstream of (or parallel to) *set-24* in mediating the TEI phenotype (similar to the statement in lines 376-377, “suggesting a role for SET-24 in ... regulation through HCF-1 ...”), a conclusion that would be a standard interpretation of epistasis. They might further juxtapose those statements with one about the observation that the two are in the same physical complex, and conclude that part of the discussion with their proposed molecular model of how the two could act antagonistically while being in a physical complex. Such clarity would help contextualize the authors’ findings and make a general statement that would highlight the discovery made.

This is a good point. We have revised the sentence as “Together, these results indicate that HCF-1 and SET-24 both mediate TEI but display opposite inheritance phenotypes, with HCF-1 acting epistatically to SET-24 in this process.” Please see lines 340-341. Furthermore, we do make explicit in the discussion that *hcf-1* could be downstream of SET-24, please see lines 511-512.

3) In lines 535 onwards, could the authors clarify their specific hypothesis about the proposed advantage of a mutant *set-24* allele in a wild isolate? A straightforward explanation would be that the Mrt phenotype is an epi-phenomenon and that there is a selective advantage under certain environments to some other consequence of reduced *set-24* function. The second hypothesis (line 547) seems to be a more complicated and perhaps unnecessary take on this general explanation. Also, the first hypothesis/explanation (line 537) could explain the Mrt “escapees”/reversibility, but I did not understand why Mrt would be beneficial in the first place? If anything, Mrt has a strong direct negative impact on fertility.

We apologise for the confusion. We did not argue that the Mrt phenotype could be beneficial. Our intention was to propose that the “escapees” may provide an explanation of how the Mrt phenotype is mitigated in wild populations. Besides that, we proposed that it may be advantageous to have such alleles in wild populations due to the generation of epigenetic variability, for example to integrate environmental stimuli in intergenerational responses. We agree that the Mrt phenotype is unlikely to be advantageous, contrary to the ability to generate epigenetic diversity. As these two

phenomena are linked consequences of set-24 mutant alleles, the negative impact on fertility could have been tolerated, partially due to epigenetic escape. Also, it is not known whether the stringent bottlenecks applied to nematode populations in the laboratory to elicit the Mrt phenotype mimic in any way the population dynamics in the wild. Therefore, it is possible that the Mrt phenotype is not as strong in the wild as in wild populations.

We have now revised this paragraph of the discussion, making adjustments throughout to clarify the overall message. Please see lines 547-557.

4) In line 306, the authors should clarify “out-of-frame” mutation. “out-of-frame” is not a standard genetic phrase and might be interpreted as “frameshift,” “deletion or insertion causing a frameshift” or other. *ok559* seems to be a deletion mutation that results in an early stop codon (and frameshift?). Whatever the nature of *ok559*, it should be clearly specified in those terms.

Thank you for the comment. We have revised “an out-of-frame mutation” to “a frameshift mutation” in line 312. The *ok559* allele is indeed an in-frame deletion within the Kelch domain¹.

Reviewer #3 (Remarks to the Author):

In this revision, the authors experimentally demonstrated that SET-24 lacks detectable HMT activity, and provided additional evidence for the interaction between SET-24 and HCF-1, as well as the domains required for this interaction. The sequencing analysis has been further strengthened and clarified. Overall, the manuscript’s conclusions are substantially stronger, and my previous concerns have been satisfactorily addressed.

Thank you for your support of our manuscript.

Reference

1. Lee, S. et al. Epigenetic regulation of histone H3 serine 10 phosphorylation status by HCF-1 proteins in *C. elegans* and mammalian cells. *PLoS One* **2**, e1213 (2007).